# Protein prenylation restrains innate immunity by inhibiting Rac1 effector interactions

Murali K. Akula [1,2], Mohamed X. Ibrahim[1,10], Emil G. Ivarsson[1,10], Omar M. Khan [3,4,10], Israiel T. Kumar[1], Malin Erlandsson[5], Christin Karlsson[1], Xiufeng Xu[6], Mikael Brisslert[5], Cord Brakebusch[7], Donghai Wang[8], Maria Bokarewa [5], Volkan I. Sayin[2,9] & Martin O. Bergo[1,6]

Rho family proteins are prenylated by geranylgeranyltransferase type I (GGTase-I), which normally target proteins to membranes for GTP-loading. However, conditional deletion of GGTase-I in mouse macrophages increases GTP-loading of Rho proteins, leading to enhanced inflammatory responses and severe rheumatoid arthritis. Here we show that heterozygous deletion of the Rho family gene *Rac1*, but not *Rhoa* and *Cdc42*, reverses inflammation and arthritis in GGTase-I-deficient mice. Non-prenylated Rac1 has a high affinity for the adaptor protein Ras GTPase-activating-like protein 1 (Iqgap1), which facilitates both GTP exchange and ubiquitination-mediated degradation of Rac1. Consistently, inactivating *Iqgap1* normalizes Rac1 GTP-loading, and reduces inflammation and arthritis in GGTase-I-deficient mice, as well as prevents statins from increasing Rac1 GTP-loading and cytokine production in macrophages. We conclude that blocking prenylation stimulates Rac1 effector interactions and unleashes proinflammatory signaling. Our results thus suggest that prenylation normally restrains innate immune responses by preventing Rac1 effector interactions.

[1] Sahlgrenska Cancer Center, Department of Molecular and Clinical Medicine, Institute of Medicine, University of Gothenburg, SE-405 30 Gothenburg, Sweden. [2] Wallenberg Centre for Molecular and Translational Medicine, University of Gothenburg, SE-405 30 Gothenburg, Sweden. [3] Adult Stem Cell Laboratory, Francis Crick Research Institute, London NW1 1AT, UK. [4] College of Health and Life Sciences, Hamad Bin Khalifa University, Education City, Qatar Foundation, Doha 34110, Qatar. [5] Department of Rheumatology, Institute of Medicine, University of Gothenburg, SE-41345 Gothenburg, Sweden. [6] Department of Biosciences and Nutrition, Karolinska Institutet, SE-141 83 Huddinge, Sweden. [7] Biotech Research and Innovation Centre, University of Copenhagen, 2200 Copenhagen N, Denmark. [8] Department of Immunology, Duke University School of Medicine, Durham, NC 27710, USA. [9] Sahlgrenska Cancer Center, Department of Surgery, Institute of Clinical Sciences, University of Gothenburg, SE-405 30 Gothenburg, Sweden. [10] These authors contributed equally: Mohamed X. Ibrahim, Emil G. Ivarsson, Omar M. Khan. Correspondence and requests for materials should be addressed to M.O.B. (email: martin.bergo@ki.se)

Protein geranylgeranyltransferase type I (GGTase-I) transfers a 20-carbon geranylgeranyl lipid to a cysteine residue of proteins harboring a carboxyl-terminal *CAAX* motif, including the Rho family proteins Rac1, RhoA, and Cdc42[1]. Geranylgeranylation, also called prenylation, enhances hydrophobicity and facilitates membrane anchoring of Rho proteins and is believed to be essential for correct subcellular targeting, effector binding, GTP loading, and activation[2,3].

Geranylgeranylation is an evolutionarily conserved modification that has generated a broad interest for several reasons. First, Rho family proteins contribute to tumor growth and metastasis, which prompted the development of GGTase-I inhibitors (GGTIs)[4]. Several GGTIs exhibit anti-tumor effects in preclinical studies and the rationale for using GGTIs in cancer therapy is supported by mouse gene-targeting experiments[5,6]. Second, Rho proteins regulate phagocytosis, migration, reactive oxygen species (ROS) production, and signaling in inflammatory cells[7]. Thus, targeting GGTase-I has been proposed as a strategy to treat inflammatory and autoimmune disorders, including rheumatoid arthritis and multiple sclerosis[8–10]. And third, reduced geranylgeranylation of Rho proteins is frequently suggested to underlie anti-inflammatory properties and other pleiotropic effects of statins[11,12]. Statins lower cholesterol levels by blocking mevalonate synthesis but this also leads to reduced synthesis of geranylgeranylpyrophosphate (GGPP), the lipid substrate of GGTase-I[13].

The idea that blocking Rho protein geranylgeranylation would inhibit inflammation was challenged by studies into GGTase-I-deficient mice[14,15]. Knockout of GGTase-I's catalytic subunit in macrophages eliminated Rho protein geranylgeranylation, but surprisingly, Rac1, RhoA, and Cdc42 accumulated in their GTP-bound active form, and Rac1 remained associated with membranes[14]. Moreover, p38 and NFκB activities were high in GGTase-I-deficient macrophages, which increased proinflammatory cytokine production after lipopolysaccharide (LPS) stimulation; and the mice developed chronic erosive rheumatoid arthritis[14]. Thus, targeting macrophage GGTase-I activates Rho proteins and causes inflammation.

These results dispute the general understanding of the biochemical and medical importance of *CAAX* protein geranylgeranylation and raise a range of new questions[16]. For example, GGTase-I has more than 60 predicted substrates[17], and it is not yet known whether one or more of these proteins mediate inflammatory signaling and erosive arthritis development in mice lacking GGTase-I in macrophages. It would also be important to define mechanisms underlying the increased GTP loading of nonprenylated Rho proteins. One potential explanation is that they interact more avidly than prenylated Rho proteins with a guanine-nucleotide exchange factor (GEF) or an adaptor protein such as Ras GTPase-activating-like protein (Iqgap1)[18–20]. Alternatively, nonprenylated Rho proteins interact less avidly with a GTPase-activating protein (GAP)[18] or guanine nucleotide dissociation inhibitor (RhoGDI1)[21]. Interestingly, knockdown of *Rhogdi1* increases GTP loading of Rho family proteins[22], and it is possible that a reduced interaction with RhoGDI1 underlies Rho-protein activation and inflammation in GGTase-I-deficient macrophages. Moreover, it would be important to establish whether statins, by reducing GGPP levels, produce similar effects in macrophages as the knockout of GGTase-I. Statin administration is frequently associated with increased Rho-GTP loading and sometimes with increased LPS-induced cytokine production, including interleukin (Il) 1β, but the underlying mechanism is unknown[23]. In the current study, we used genetic, pharmacologic, and proteomic strategies to address those issues and identify a new potential explanation for the role of prenylation for Rac1 effector interactions and proinflammatory signaling.

## Results

### Rac1 knockout prevents arthritis in GGTase-I-deficient mice.

Mice lacking GGTase-I in macrophages ($Pggt1b^{fl/fl}$LysM-$Cre^{+/0}$, hereafter designated $Pggt1b^{\Delta/\Delta}$) develop erosive arthritis, and nonprenylated Rac1, RhoA, and Cdc42 accumulate in the GTP-bound state[14]. In TX-114 phase-separation assays Rac1, RhoA, and Cdc42 exhibited a substantial shift from the detergent to aqueous phase in $Pggt1b^{\Delta/\Delta}$ macrophages; and click-chemistry experiments revealed that Rac1 is not geranylgeranylated—thus confirming the previous conclusion that these proteins are not prenylated (Supplementary Fig. 1A and B)[14]. We hypothesized that the increased activity of one of these nonprenylated proteins mediates arthritis development and the proinflammatory phenotypes of $Pggt1b^{\Delta/\Delta}$ mice. To test this hypothesis, we knocked out one copy of *Rac1*, *Rhoa*, and *Cdc42* in macrophages of $Pggt1b^{\Delta/\Delta}$ mice (previous studies show that knockout of both copies of *Rac1*, *Rhoa*, and *Cdc42* produces multiple in vivo and cellular phenotypes)[24–29]. As expected, levels of total Rac1, RhoA, and Cdc42 were ~50% (47–58%) lower in BM macrophages from $Pggt1b^{\Delta/\Delta}Rac1^{\Delta/+}$, $Pggt1b^{\Delta/\Delta}Rhoa^{\Delta/+}$, and $Pggt1b^{\Delta/\Delta}Cdc42^{\Delta/+}$ mice, respectively, than in macrophages from littermate $Pggt1b^{\Delta/\Delta}$ mice (Supplementary Fig. 1C). Similarly, levels of GTP-bound Rac1 was ~50% lower in $Pggt1b^{\Delta/\Delta}Rac1^{\Delta/+}$ than in $Pggt1b^{\Delta/\Delta}$ macrophages (Fig. 1a). Immunohistochemical analyses revealed that the high synovitis and bone erosion scores in joints of $Pggt1b^{\Delta/\Delta}$ mice were markedly lower in $Pggt1b^{\Delta/\Delta}Rac1^{\Delta/+}$ mice, and were statistically indistinguishable from wild type (Fig. 1b). The arthritis scores were similar in $Pggt1b^{\Delta/\Delta}Rhoa^{\Delta/+}$ and $Pggt1b^{\Delta/\Delta}$ mice whereas they were higher in $Pggt1b^{\Delta/\Delta}Cdc42^{\Delta/+}$ mice (Supplementary Fig. 1D and E).

As in earlier studies, LPS-stimulated $Pggt1b^{\Delta/\Delta}$ macrophages produced and secreted Il-1β (which wild-type macrophages do not); this effect was associated with caspase-1 activation (Fig. 1c, d). Inactivation of *Rac1* reduced caspase-1 and Il-1β production by 90–100% (Fig. 1c, d). LPS-stimulated $Pggt1b^{\Delta/\Delta}$ macrophages also secreted high amounts of Il-6, Tnf, and Mmp13; whereas the levels in medium of $Pggt1b^{\Delta/\Delta}Rac1^{\Delta/+}$ and wild-type macrophages were similar (Fig. 1c, e). In addition, basal expression of inflammation and extracellular matrix-associated genes were increased in $Pggt1b^{\Delta/\Delta}$ macrophages, but similar to wild-type in $Pggt1b^{\Delta/\Delta}Rac1^{\Delta/+}$ cells (Supplementary Fig. 1F and G). Cytokine levels in medium of $Pggt1b^{\Delta/\Delta}Rhoa^{\Delta/+}$ and $Pggt1b^{\Delta/\Delta}Cdc42^{\Delta/+}$ macrophages did not differ consistently compared with $Pggt1b^{\Delta/\Delta}$ (Supplementary Fig. 2A–D).

$Pggt1b^{\Delta/\Delta}$ macrophages exhibited increased levels of LPS-stimulated p38 phosphorylation and phosphorylation of the Nf-κB-regulator IκB kinase (Ikk); and increased basal and LPS-stimulated phosphorylation of the tyrosine protein kinase Src and signal transducer and activator of transcription 3 (Stat3) (Fig. 1f). Consistent with results in Fig. 1a–e, phosphorylation of p38, Src, Ikk, and Stat3 was reduced in $Pggt1b^{\Delta/\Delta}Rac1^{\Delta/+}$ macrophages (Fig. 1f).

Rac1-GTP may activate inflammatory signaling pathways through multiple effectors including ROS and p38. Pharmacological inhibition of ROS-reduced phosphorylation of Stat3, Src, and Ikk in $Pggt1b^{\Delta/\Delta}$ macrophages, but did not influence p38 phosphorylation (Supplementary Fig. 2E and F). Moreover, a p38 inhibitor and a ROS inhibitor prevented the increased cytokine production observed in LPS-stimulated $Pggt1b^{\Delta/\Delta}$ macrophages (Fig. 1g and Supplementary Fig. 2G and H).

### GGTase-I knockout increases Rac1-GTP, but reduces total Rac1.

Nonprenylated Rac1, RhoA, and Cdc42 in $Pggt1b^{\Delta/\Delta}$ macrophages accumulate in their GTP-bound active forms[14] (Fig. 2a and Supplementary Fig. 3A and B); gene-expression levels

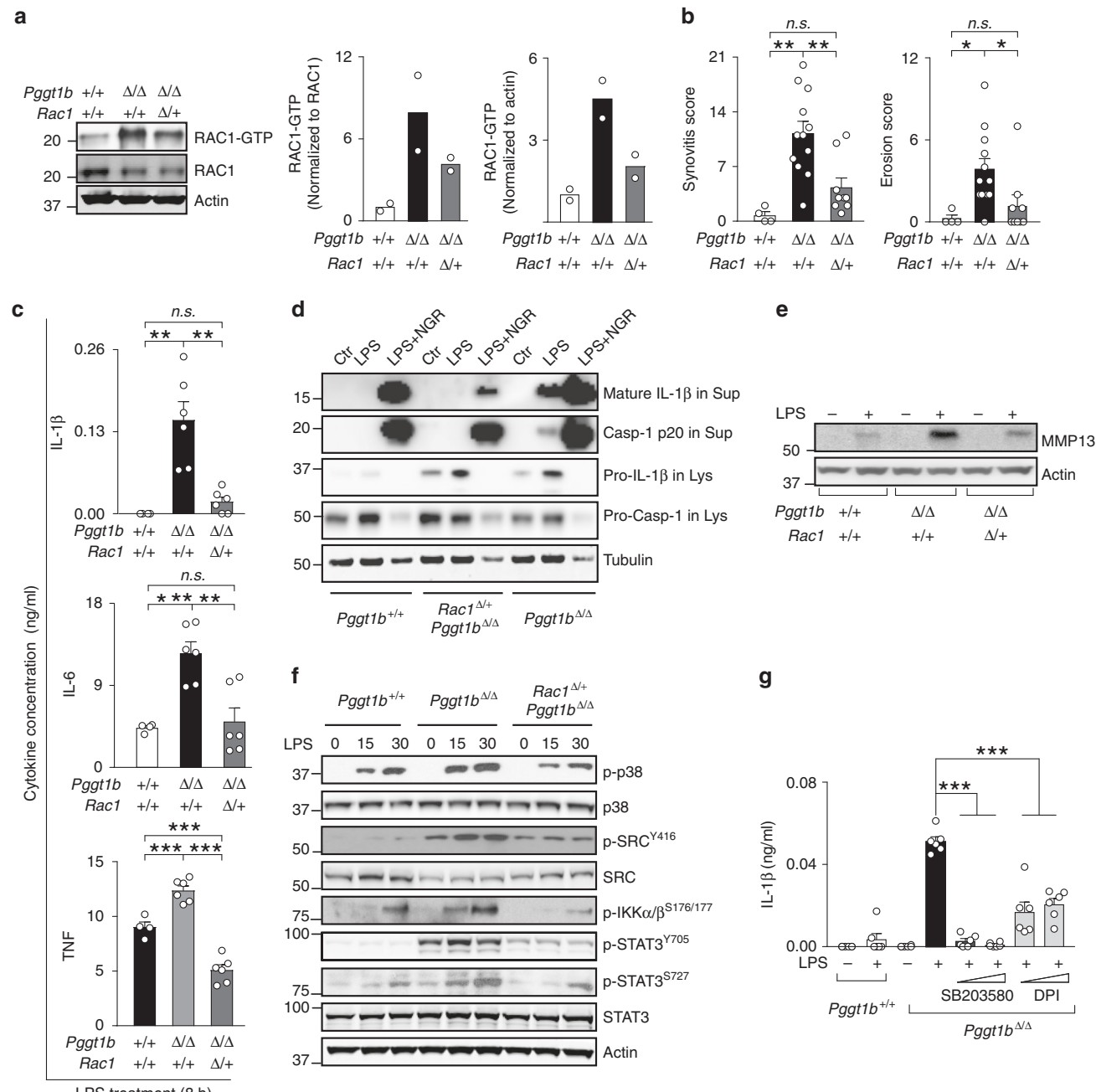

**Fig. 1** *Rac1* haploinsufficiency rescues arthritis and inflammatory signaling in *Pggt1b*$^{\Delta/\Delta}$ mice. **a** Left, Western blots showing steady-state levels of GTP-bound and total Rac1 in BM macrophages isolated from *Pggt1b*$^{\Delta/+}$, *Pggt1b*$^{\Delta/\Delta}$, and littermate *Rac1*$^{\Delta/+}$*Pggt1b*$^{\Delta/\Delta}$ mice. Actin was used as a loading control. Right, Bar graphs showing mean Rac1-GTP levels determined by densitometry ($n = 2$ per genotype). **b** Synovitis and erosion score in joints of 12-week-old *Pggt1b*$^{+/+}$ ($n = 4$), *Pggt1b*$^{\Delta/\Delta}$ ($n = 12$), and *Rac1*$^{\Delta/+}$*Pggt1b*$^{\Delta/\Delta}$ ($n = 9$) mice. **c** Cytokine concentrations, 8 h after LPS (10 ng/ml) stimulation, in medium of primary bone marrow (BM) macrophages isolated from *Pggt1b*$^{+/+}$ ($n = 3$), *Pggt1b*$^{\Delta/\Delta}$ ($n = 4$), and *Rac1*$^{\Delta/+}$*Pggt1b*$^{\Delta/\Delta}$ ($n = 3$) mice. **d** Western blots showing levels of mature Il-1β and caspase-1 in supernatants (Sup), and pro-Il-1β and pro-caspase-1 in lysates (Lys) of LPS (200 ng/ml) stimulated BM macrophages; tubulin in lysates was used as a loading control. The antibiotic nigericin (28 mM) was used as a positive control for inflammasome-mediated caspase-1 activation and Il-1β production. **e** Western blot showing levels of Mmp13 in medium of LPS-stimulated BM macrophages; Actin in lysates was used as a loading control. **f** Western blots showing phosphorylated (p) and total levels of intracellular signaling mediators in lysates of BM macrophages isolated 0, 15, and 30 min after LPS stimulation. **g** Concentration of Il-1β in medium of LPS-stimulated BM macrophages ($n = 3$/genotype) that had been pre-incubated for 1 h with inhibitors of p38 (SB203580; 1 and 5 μM) and ROS (DPI; 500 nM and 5 μM). For **c**–**e**, **g**, similar results were observed in two to three independent experiments. Error bars presented as s.e.m. when $n$ is equal to or more than three. Significance between groups were calculated with two tailed Student's *t* test (**c**, **g**) and one-way ANOVA with Tukey's post hoc test (**b**). n.s. not significant, $*P < 0.05$, $**P < 0.01$, $***P < 0.001$

were unaffected (Fig. 2b). Further analyses revealed that the total levels of Rac1 were reduced by ~50% (Fig. 2a). Moreover, the half-life of Rac1 was 65–70% shorter in *Pggt1b*$^{\Delta/\Delta}$ than in control cells (Fig. 2c, d). To determine whether ubiquitin-mediated

proteasomal degradation contributed to the reduced Rac1 levels, we first immunoprecipitated ubiquitin and performed western blots for Rac1; and found higher amounts of ubiquitin-associated Rac1 in *Pggt1b*$^{\Delta/\Delta}$ than in control lysates (Fig. 2e). Although we

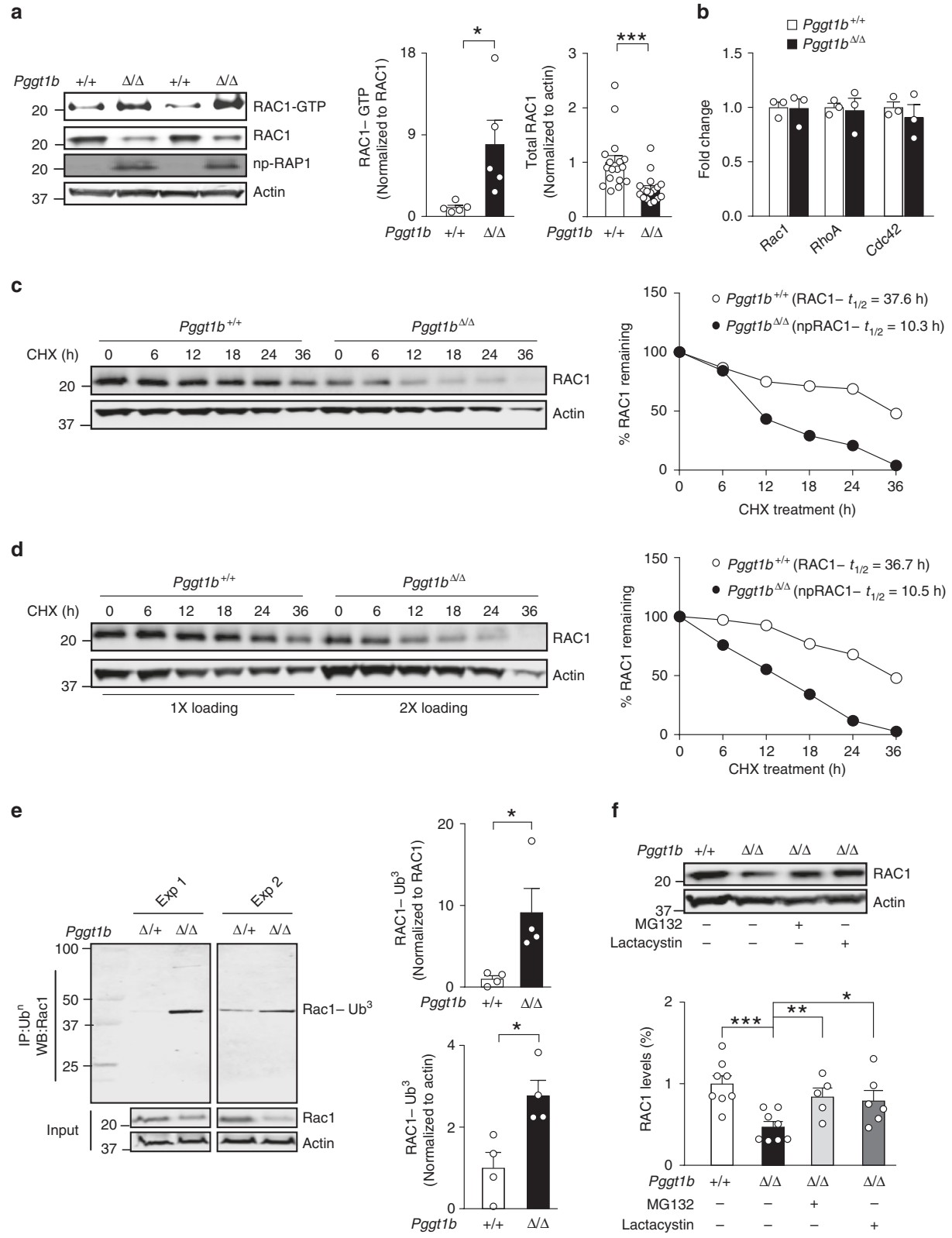

only detected Rac1 conjugated with three Ubiquitins under the current experimental conditions, we can't rule out the existence of mono-, di-, and polyubiquitinated forms. Second, we incubated macrophages with proteasome inhibitors and found that total Rac1 levels were restored (Fig. 2f); proteasome inhibition also increased Rac1-GTP levels (Supplementary Fig. 3C). In contrast with Rac1, total RhoA, and Cdc42 levels were increased in *Pggt1b*^Δ/Δ cells (Supplementary Fig. 3A and B). Thus, blocking prenylation reduces Rac1 stability but appears to increase that of RhoA and Cdc42.

**Fig. 2** GGTase-I knockout increases Rac1-GTP loading, but reduces Rac1 total levels. **a** Left, western blots showing steady-state levels of GTP-bound and total Rac1 in BM macrophages. Nonprenylated RAP1A was used as marker of GGTase-I-deficient cells; actin was used as a loading control. Middle and right, amount of GTP-bound ($n = 5$/genotype) and total Rac1 ($n = 17$/genotype) in BM macrophages determined by densitometry of protein bands. **b** Quantitative polymerase chain reaction (QPCR) data showing levels of *Rac1*, *Rhoa*, and *Cdc42* expression in cDNA of BM macrophages ($n = 3$ per genotype). **c** Left, western blots showing levels of Rac1 and Actin that remain in BM macrophages at various time points after incubation with cycloheximide (20 μg/ml) to stop protein synthesis. Equal amounts of total proteins were loaded. Right, densitometry of protein bands normalized to time-point 0 within each genotype. **d** Similar experiment as in (**c**) except twice the amount of total proteins from the *Pggt1b*$^{Δ/Δ}$ lysates were loaded compared to *Pggt1b*$^{Δ/+}$ to obtain similar Rac1 levels at time-point 0. **e** Left, immunoprecipitation (IP) of Ubiquitin (Ub) followed by western blots for Rac1. Direct western blots were performed on the same lysates (input) to quantify total levels of Rac1 and Actin. The molecular weight of the main band was ~45 kDa which corresponds to Rac1 conjugated with three Ubs. Two independent experiments are shown. Right, amount of ubiquitin-bound Rac1 determined by densitometry of protein bands ($n = 4$/genotype). **f** Upper panel, western blots showing total Rac1 levels in lysates of BM macrophages incubated for 10 h with proteasome inhibitors MG-132 (15 μM) and lactacystin (15 μM). Lower panel, quantification of protein bands normalized to Actin and expressed as percent of control (*Pggt1b*$^{Δ/+}$). Similar results were obtained three times. Error bars represent s.e.m. Significance between groups were calculated with two tailed Student's *t* test. *$P < 0.05$, **$P < 0.01$, ***$P < 0.001$

**Table 1 Rho family effector proteins identified by mass spectrometry of proteins co-immunoprecipitated with Rac1 in *Pggt1b*$^{Δ/Δ}$ and control macrophages**

| S No. | Accession | Description | PSM | Fold change | *P* value |
|---|---|---|---|---|---|
| 1 | Q9JKF1 | Ras GTPase-activating-like protein Iqgap1 [Iqgap1] | 98 | 1.16 | 0.007 |
| 2 | Q640N3 | Rho GTPase-activating protein 30 [RHG30] | 5 | 1.12 | 0.012 |
| 3 | Q8BYW1 | Rho GTPase-activating protein 25 [RHG25] | 2 | 1.60 | 0.043 |
| 4 | A6X8Z5 | Rho GTPase-activating protein 31 [RHG31] | 9 | 0.85 | 0.170 |
| 5 | Q9WVM1 | Rac GTPase-activating protein 1 [RGAP1] | 3 | 0.84 | 0.227 |

Fold change represents the relative abundance of a protein in macrophages from the two genotypes.

**RhoGDI1 is not involved in phenotypes of GGTase-I deficiency**. The GGTase-I knockout reduces interactions between RhoGDI1 and Rac1 and Cdc42, but not RhoA[14]. We hypothesized that a reduced interaction between RhoGDI1 and Rac1 might underlie increased GTP loading and cytokine production and tested this by inactivating RhoGDI1 expression. Consistent with a previous study[22], suppressing RhoGDI1 expression with small-interfering (si) RNAs increased Rac1 GTP loading and reduced total Rac1 levels—a result that was associated with increased LPS-stimulated Il-6 and Tnf production (Supplementary Fig. 4A). However, we only observed this result in the RAW264.7 macrophage cell line; it was neither observed in primary macrophages incubated with siRNAs, nor in immortalized macrophages where the *Arhgdia1* gene had been inactivated with CRISPR/CAS9 (Supplementary Fig. 4B and C). Moreover, RhoGDI1 inactivation never increased Il-1β production, including in RAW264.7 cells (Supplementary Fig. 4A–C).

**Iqgap1 binds nonprenylated Rac1 and mediates arthritis**. To determine whether blocking Rac1 prenylation influences other effector interactions, we immunoprecipitated Rac1 from *Pggt1b*$^{Δ/Δ}$ and control macrophage lysates and performed isobaric tagging for relative peptide quantification with mass spectrometry[30]. From a list of 717 proteins whose levels differed significantly in *Pggt1b*$^{Δ/Δ}$ and control cells, we identified five known Rho family effector proteins (Table 1). The top hit was GTPase-activating-like protein 1 (Iqgap1)—an adaptor protein that binds and stabilizes GTP-bound Rho family proteins but lacks GAP and GEF activity[20,31,32]. Immunoprecipitation (IP) and western blot analyses of macrophage lysates revealed that the Rac1-Iqgap1 interaction was two to threefold higher in *Pggt1b*$^{Δ/Δ}$ than control cells (Fig. 3a). The interaction between Iqgap1 and RhoA and Cdc42 in *Pggt1b*$^{Δ/Δ}$ macrophages was also increased (Supplementary Fig. 5A and B).

To determine whether Iqgap1 contributes to phenotypes of GGTase-I deficiency, we bred *Pggt1b*$^{Δ/Δ}$ mice on an *Iqgap1*$^{-/-}$ background[33]. Levels of synovitis and bone erosion were 60–70% lower in joints of *Pggt1b*$^{Δ/Δ}$*Iqgap1*$^{-/-}$ than *Pggt1b*$^{Δ/Δ}$*Iqgap1*$^{+/+}$ mice, and cytokine production by LPS-stimulated *Pggt1b*$^{Δ/Δ}$*Iqgap1*$^{-/-}$ macrophages was reduced to control levels (Fig. 3b, c). Importantly, knockout of *Iqgap1* alone (e.g., *Pggt1b*$^{+/+}$ *Iqgap1*$^{-/-}$) did not influence cytokine production (Supplementary Fig. 5C), although it reduced basal Rac1-GTP and total Rac1 levels (Supplementary Fig. 5D). Moreover, the *Iqgap1* knockout reduced Rac1-GTP loading and ubiquitination, and increased total Rac1 to levels observed in controls; and essentially normalized LPS-induced p38, Src, and Stat3 phosphorylation (Fig. 3d, e, and Supplementary Fig. 5E). The *Iqgap1* knockout also reduced GTP-bound and total RhoA and Cdc42 to control levels (Supplementary Fig. 5F and G). Furthermore, *Pggt1b*$^{Δ/Δ}$ macrophages have a small adhesive area and appear small and rounded, and the knockout of *Iqgap1* abolished this phenotype (Fig. 3f).

**Tiam1 binds nonprenylated Rac1 and stimulates GTP loading**. Iqgap1 has no GEF or GAP activity[20,31]. Thus, we asked whether an increased interaction with a known GEF—or a reduced interaction with RacGAP1—could explain the increased GTP loading of non-prenylated Rac1. IP-western blot analyses revealed a two to sixfold higher Rac1-Tiam1 interaction in *Pggt1b*$^{Δ/Δ}$ than in control macrophages (Fig. 4a). Interactions between Rac1 and the GEFs Vav1, Vav2, β-Pix, and Dock1, and the GAP RacGAP1 were similar in *Pggt1b*$^{Δ/Δ}$ and control macrophages (Supplementary Fig. 6A). Inhibiting Tiam1 expression with siRNAs reduced GTP-bound and increased total Rac1 (Fig. 4b). Tiam1 inhibition also reduced cytokine production of LPS-stimulated *Pggt1b*$^{Δ/Δ}$ macrophages (Fig. 4c and Supplementary Fig. 6B and C).

To determine whether Tiam1 interacts with Iqgap1, we performed IP-western blot analyses and found a low basal interaction in control *Pggt1b*$^{+/+}$ macrophages and a high degree of interaction in *Pggt1b*$^{Δ/Δ}$ macrophages (Fig. 4d). These experiments also revealed consistently higher Tiam1 levels in

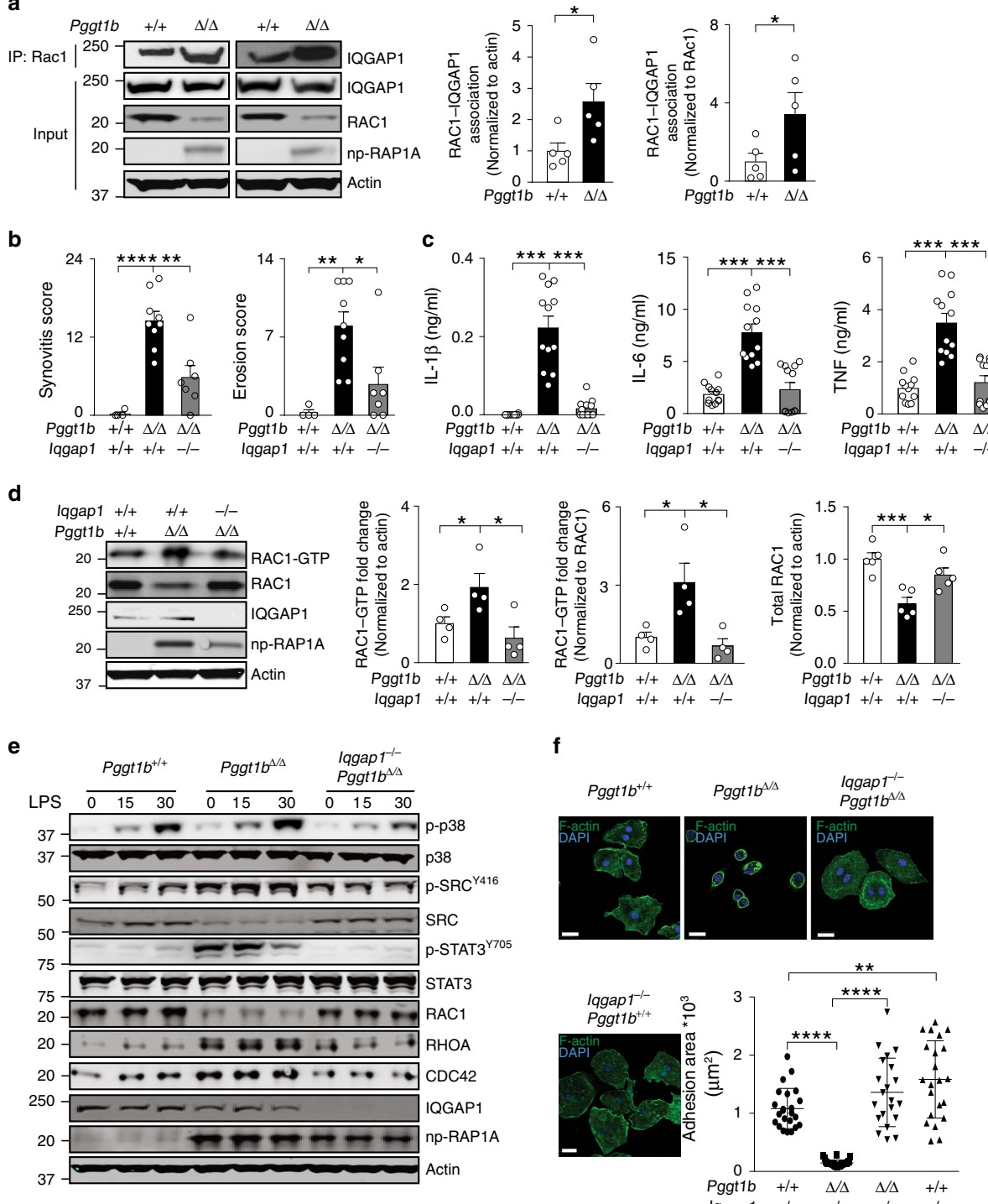

**Fig. 3** Iqgap1 binds nonprenylated Rac1 and mediates GTP loading, cytokine production, and arthritis. **a** Left, immunoprecipitation (IP) of Rac1 in BM macrophage lysates followed by western blots for Iqgap1. Direct western blots were performed on the same lysates (Input) to quantify total Iqgap1 and Rac1 levels. Nonprenylated Rap1A was used as marker of GGTase-I-deficient cells; actin was used as a loading control. Middle and right, levels of Rac1-bound Iqgap1 was determined by densitometry of IP-western blots of BM macrophages from six mice/genotype. **b** Synovitis and erosion scores in joints of 12-week-old $Pggt1b^{\Delta/+}$ ($n = 4$), $Pggt1b^{\Delta/\Delta}$ ($n = 9$), and littermate $Iqgap1^{-/-}Pggt1b^{\Delta/\Delta}$ ($n = 7$) mice. **c** Cytokine concentrations, 8 h after LPS stimulation, in medium of BM macrophages isolated from $Pggt1b^{\Delta/+}$, $Pggt1b^{\Delta/\Delta}$, and $Iqgap1^{-/-}Pggt1b^{\Delta/\Delta}$ mice ($n = 3$/genotype). **d** Left panel, Western blots showing levels of GTP-bound Rac1, total Rac1, and Iqgap1 in BM macrophages isolated from $Pggt1b^{\Delta/+}$, $Pggt1b^{\Delta/\Delta}$, and $Iqgap1^{-/-}Pggt1b^{\Delta/\Delta}$ mice. Right panels, levels of GTP-bound and total Rac1 determined by densitometry ($n = 5$/genotype). **e** Western blots of macrophage lysates isolated at base-line and 15 and 30 min after LPS stimulation. **f** Confocal microscopy images showing F-Actin staining of BM macrophages. Scale bars: 10 μm. Error bars represent s.e.m. The graph shows adhesive area of 20–30 macrophages in ten randomly selected fields analyzed in cells from two mice/genotype. Significance between groups were calculated with two tailed Student's $t$ test (**a**, **c**, **d**, **f**) and one-way ANOVA with Tukey's post hoc test (**b**). n.s. not significant, $*P < 0.05$, $**P < 0.01$, $***P < 0.001$

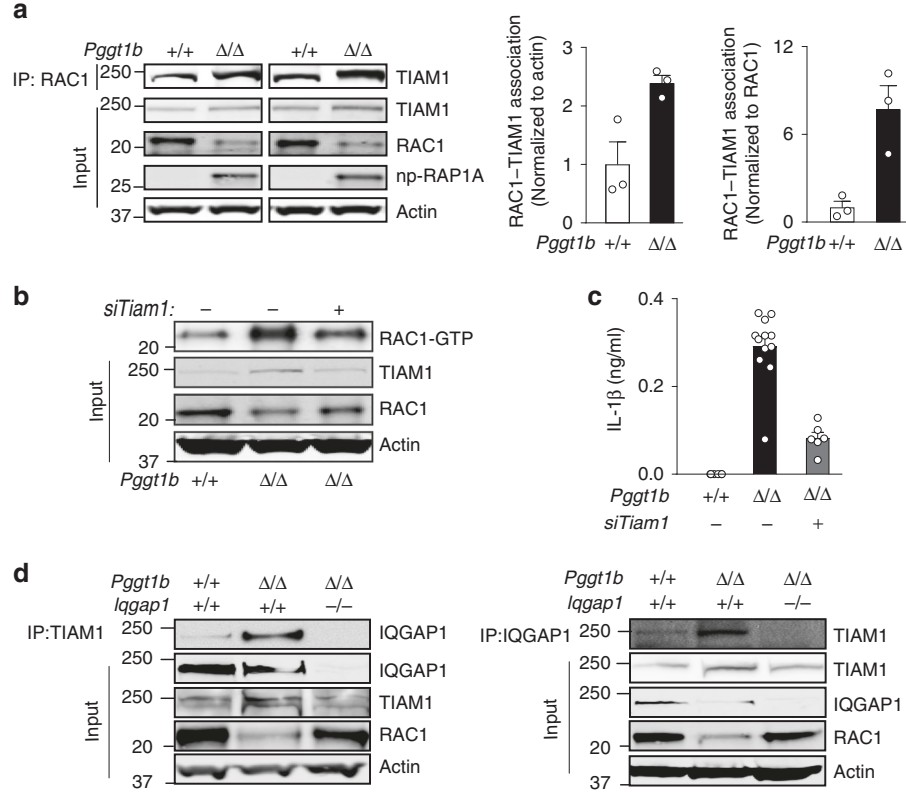

**Fig. 4** Tiam1 binds nonprenylated Rac1 and supports GTP loading and Il-1β production. **a** Left, immunoprecipitation (IP) of Rac1 in BM macrophage lysates followed by western blots for Tiam1. Direct western blots were performed on the same lysates (Input) to quantify total Tiam1 and Rac1 levels. Middle and right, levels of Rac1-bound Tiam1 was determined by densitometry of IP-western blots ($n = 3$/genotype). Actin was the loading control. **b** Western blot showing levels of GTP-bound Rac1 in lysates of BM macrophages pre-incubated for 24 h with scrambled or *Tiam1*-targeted siRNAs. Direct western blots were performed on the same lysates (Input) to quantify total Tiam1 and Rac1 levels. **c** Concentration of Il-1β, 8 h after LPS stimulation, in medium of BM macrophages pre-incubated for 24 h with scrambled or *Tiam1*-targeted siRNAs ($n = 2$ genotype). **d** Left, IP of Tiam1 followed by western blot for Iqgap1. Right, IP of Iqgap1 followed by western blot for Tiam1. Direct western blots were performed on the same lysates (Input) to quantify total Iqgap1, Tiam1, and Actin levels. Error bars represent s.e.m. Significance between groups were calculated with two tailed Student's $t$ test. *$P < 0.05$

*Pggt1b*$^{Δ/Δ}$ than in control macrophages (Fig. 4a, b, d); the *Iqgap1* knockout normalized Tiam1 levels (Fig. 4d).

**Editing the Rac1 *CAAX* sequence increases GTP loading.** Levels of GTP-bound and total Rac1 in GGTase-I-deficient cells could conceivably be influenced by the accumulation of other nonprenylated *CAAX*-proteins. To define biochemical consequences of blocking Rac1 prenylation in cells expressing normal GGTase-I activity, we edited the *CAAX* sequence of endogenous *Rac1* in human embryonic kidney (HEK) cells with a nonintegrating homologous recombination-based CRISPR/CAS9 approach. We isolated three *CAAX*-mutant clones (*Rac1CM1–3*), sequenced their DNA and cDNA (Fig. 5a), and found in their lysates that Rac1 exhibited a reduced electrophoretic mobility indicating that the protein had not been prenylated by GGTase-I (Fig. 5b). Rac1-GTP levels were two to fourfold higher in *CAAX*-mutant than in control cells; and Rac1 total levels were two to threefold lower (Fig. 5c). Further analyses revealed that the distribution of Rac1 in cytosol and membrane fractions was similar in *CAAX*-mutant and control cells (Fig. 5d). However, nuclear Rac1 levels were lower in the edited cells (Fig. 5d). Similar to the findings with *Pggt1b*$^{Δ/Δ}$ macrophages, Rac1 total levels increased following lactacystin and MG-132 administration (Fig. 5e, f). Moreover, Rac1 ubiquitination and Iqgap1 association was higher in Rac1 mutant than control cells (Fig. 5g, h).

**Statins mimic effects of GGTase-I deficiency.** To determine whether statins can produce similar cellular effects as GGTase-I deficiency, we incubated three different macrophage types with Atorvastatin, Rosuvastatin, and Simvastatin. Statins increased Rac1-GTP levels and reduced total Rac1; and increased GTP-bound and total RhoA (Fig. 6a, b). Moreover, statins facilitated Il-1β maturation and secretion in response to LPS, and potentiated Il-6 and Tnf production (Fig. 6c, d, and Supplementary Fig. 7A–C). Pre-incubating the macrophages with GGPP abolished the statin effect on cytokine production in most, but not all cases (Fig. 6c, d, and Supplementary Fig. 7A and B). To determine if Iqgap1 underlies the ability of statins to increase cytokine production, we incubated primary *Iqgap1*-knockout macrophages with Simvastatin and found that LPS-stimulated cytokine production was either reduced or normalized (Fig. 6e). Inhibiting Iqgap1 with siRNAs produced similar results (Supplementary Fig. 7D).

## Discussion

In this study, we identified three components of the mechanism underlying inflammatory phenotypes of mice with macrophage-specific GGTase-I deficiency. First, GGTase-I prenylates at least 60 substrates[17], and our data indicate that one of them, Rac1, mediates the majority of the robust innate immune responses. Second, nonprenylated Rac1 becomes hyperactivated through an

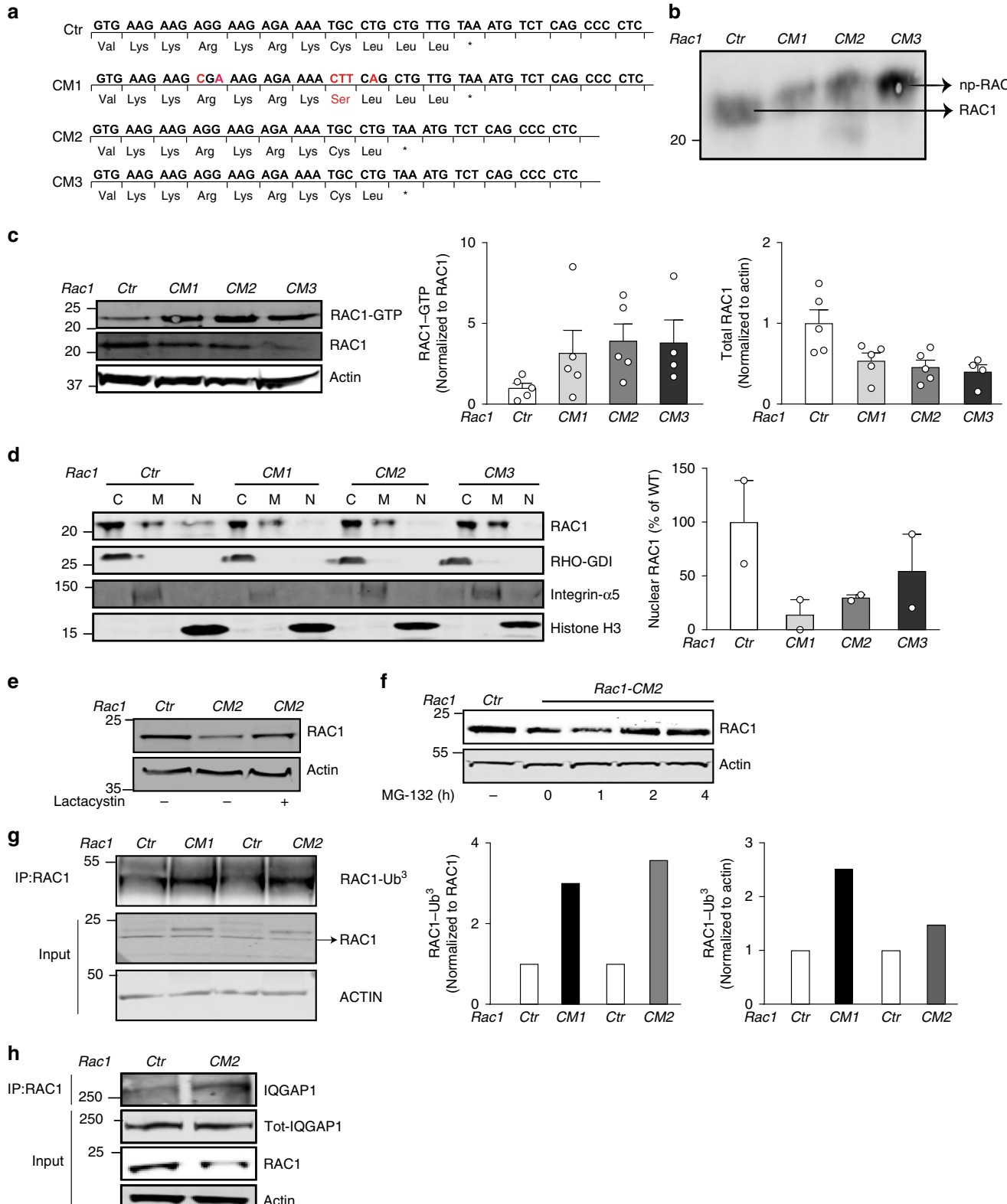

increased interaction with Tiam1 and Iqgap1. And third, hyperactive Rac1 activates inflammasome-, ROS-, and p38-driven signaling pathways that enhance LPS-induced Il-1β, Il-6, and Tnf production.

Prenylation is believed to have evolved as a strategy to target proteins to membranes, promote effector interactions, and thereby stimulate activation and signaling[34]. Several results

indicate that this is not the case for Rac1. First, Rac1 membrane/cytosol partitioning was unaffected in GGTase-I-deficient macrophages[14] and in HEK cells engineered to express endogenous Rac1 with *CAAX* motif mutations. Second, interactions between nonprenylated Rac1 and the effectors Tiam1 and Iqgap1 were substantially increased. And third, GTP-binding was increased and downstream signaling pathways were activated. The simplest

**Fig. 5** Editing the endogenous Rac1 *CAAX*-motif increases GTP loading and reduces total Rac1 levels. **a** Predicted amino acid sequence from sanger sequence results of PCR-amplified *Rac1* DNA and cDNA fragments. The DNA/cDNA was from HEK-293 clones whose *Rac1* gene had been edited at the *CAAX* sequence by CRISPR/Cas9-facilitated homologous recombination to prevent geranylgeranylation by GGTase-I. CM1–3, *CAAX* mutant clones; *Ctr*, un-edited control clone. **b** Left, western blot showing electrophoretic mobility of wild-type Rac1 in *Rac1Ctr* lysates, and nonprenylated (np) Rac1 in *CM1–3* lysates. **c** Left, western blots showing amounts of GTP-bound Rac1, total Rac1, and the loading control Actin in lysates of the gene-edited cells. Middle and right, levels of GTP-bound and total Rac1 determined by densitometry; data are mean of three independent experiments for each cell line. **d** Left, western blots showing the distribution of Rac1 in cytosol, membrane, and nuclear fractions (designated C, M, and N, respectively) of the gene-edited cells. Rho-GDI1, integrin-α5, and histone H3 were used as markers for cytosol, membrane, and nuclear fractions, respectively. Right, levels of nuclear Rac1 determined by densitometry data of three independent experiments and expressed as percent of *Rac1Ctr*. **e, f** Western blots showing levels of total Rac1 and the loading control Actin in lysates of gene-edited cells isolated after incubation with 15 μM lactacystin for 12 h (**e**); and 50 μM MG-132 at different time points (**f**). **g** Left, immunoprecipitation (IP) of Rac1 in gene-edited cells followed by western blot for Ubiquitin. Direct western blots were performed on the same lysates (Input) to quantify total Rac1 levels. Actin was used as a loading control. Right, Bar graphs showing levels of Rac1-Ub[3]. **h** Immunoprecipitation (IP) of Rac1 in gene-edited cells followed by western blot for Iqgap1. Direct western blots were performed on the same lysates (Input) to quantify Iqgap1 and Rac1 levels. Actin was used as a loading control. Error bars represent s.e.m

explanation for these results is that GGTase-I-mediated pre-nylation normally acts as a break on innate immune responses in macrophages by limiting Rac1 effector interactions. Blocking prenylation releases the break, stimulates Rac1 interactions, and unleashes wide-spread proinflammatory signaling (Supplementary Fig. 8).

This reasoning is particularly relevant for the control of Il-1β production. Due to its potent pleiotropic inflammatory effects, Il-1β secretion by macrophages is tightly controlled and requires two different events: a priming step through cell-surface receptors that leads to *Il1b* transcription; and a second signal that activates caspase-1 and leads to pro-Il-1β cleavage and secretion of the mature protein[35,36]. Our data suggest that nonprenylated Rac1 can act as the second signal in Il-1β production. This argument is supported by the finding that the robust caspase-1-mediated Il-1β production in GGTase-I-deficient macrophages was abolished by normalizing Rac1-GTP levels—which was accomplished by knocking out one copy of *Rac1* or both copies of *Iqgap1*.

GGTase-I deficiency increased basal phosphorylation of Src, Stat3, and Ikkα/β—and to some extent p38—in a Rac1-dependent fashion; LPS-induced phosphorylation of all four proteins was also increased. Rac1-induced activation of Src, Stat3, and Ikk was mediated by ROS, whereas p38 activation was likely a direct consequence of Rac1/PAK activity. These findings are consistent with previous reports that Rac1 triggers ROS production by NADPH oxidases which activates Stat3[37,38]; although other studies show a physical interaction between Rac1-GTP and Stat3[39]. Interestingly, Stat3 contributes to the progression of chronic inflammation and joint destruction in mouse models of rheumatoid arthritis[40,41]. Src and p38 are also involved in Ikk-dependent activation of NFκB which stimulates transcription of cytokines including Il-1β, Il-6, and Tnf[42–44]. Importantly, knockout of one copy of *Rac1* normalized Rac1-GTP levels in GGTase-I-deficient macrophages and reduced or normalized signaling of the entire pathway.

Knockout of *Iqgap1* normalized Rac1-GTP loading, abolished proinflammatory signaling and cytokine production, and mark-edly reduced arthritis in GGTase-I-deficient mice. Although it is not known whether Iqgap1 encounters nonprenylated Rho proteins in activated wild-type macrophages, it is possible that targeting Iqgap1 might be useful in the therapy of some inflammatory conditions. One example would be patients with mevalonate kinase deficiency (MKD)—an autoinflammatory disease associated with high Il-1β production, fever episodes, lymph node enlargment, and joint pain. MKD leads to reduced synthesis of GGPP which reduces prenylation[23,45–47], and could therefore enhance cytokine production through an increased Rac1-Iqgap1 interaction. Targeting Iqgap1 would likely be associated with few side-effects as *Iqgap1*-deficient mice are viable and

exhibit only mild phenotypes late in life[33]; and their macrophages responded normally to LPS. Targeting Rac1 might be an alternative strategy and has been proposed earlier[23,46]; but *Rac1* deficiency is lethal in mice and produces a range of cellular and tissue phenotypes.

The knockout of *Iqgap1* rescued most of the robust proinflammatory phenotypes of GGTase-I-deficient macrophages but did not influence LPS-induced cytokine production of GGTase-I wild-type macrophages—despite reducing Rac1-GTP levels. The simplest explanation for these observations is that both the levels of Rac1-GTP and the affinity of Rac1 for Iqgap1 were markedly higher in GGTase-deficient than wild-type macrophages; thus Iqgap1's role in controlling Rac1-GTP levels and its downstream signaling could simply be comparatively more important in the GGTase-I-deficient cells. Whether or not Iqgap1 influences Rac1-GTP signaling and cytokine production in arthritis and other inflammatory diseases in the setting of wild-type GGTase-I remains to be determined.

Statin administration produced similar effects as the knockout of GGTase-I: Rac1-GTP levels increased, as did LPS-induced cytokine production, in a GGPP-dependent fashion. Part of those results are in line with previous studies[23,48], but here we also provide evidence that the proinflammatory statin effect requires Iqgap1. It, therefore, seems reasonable to speculate that statins' pro-inflammatory effects on macrophages stem from reduced Rac1 prenylation and increased interaction with Iqgap1. However, statin therapy is more often associated with anti-inflammatory than pro-inflammatory effects. A potential explanation is that anti-inflammatory statin effects are the result of the drug's action on lymphocytes rather than macrophages. For example, statins inhibit T-cell proliferation and differentiation, and immune synapse formation[12]. Another speculation would be that some side-effects of statin therapy, such as myositis and rhabdomyolysis, are caused by hyperactivation of a nonprenylated Rho protein. However, these speculations should be interpreted with caution because there is little evidence that statins inhibit prenylation in vivo. Daily statin doses used by patients range from 5 to 80 mg/day, resulting in plasma concentrations of 1–15 nM[49]. The doses used in vitro in the present study (i.e., 1–5 μM) are lower than those of many other studies[50–52], but they are likely higher than those that cells in vivo are exposed to.

Our results show clearly that prenylation is not required for GTP loading and activation of Rho proteins. Prenylation actually inhibits GTP loading. Why then are Rho proteins prenylated? One potential role of prenylation is to fine-tune the targeting of Rho proteins to specific subcellular membrane domains. Another possibility is that prenylation is required for rapid cycling between GTP and GDP-bound states during specific functions of

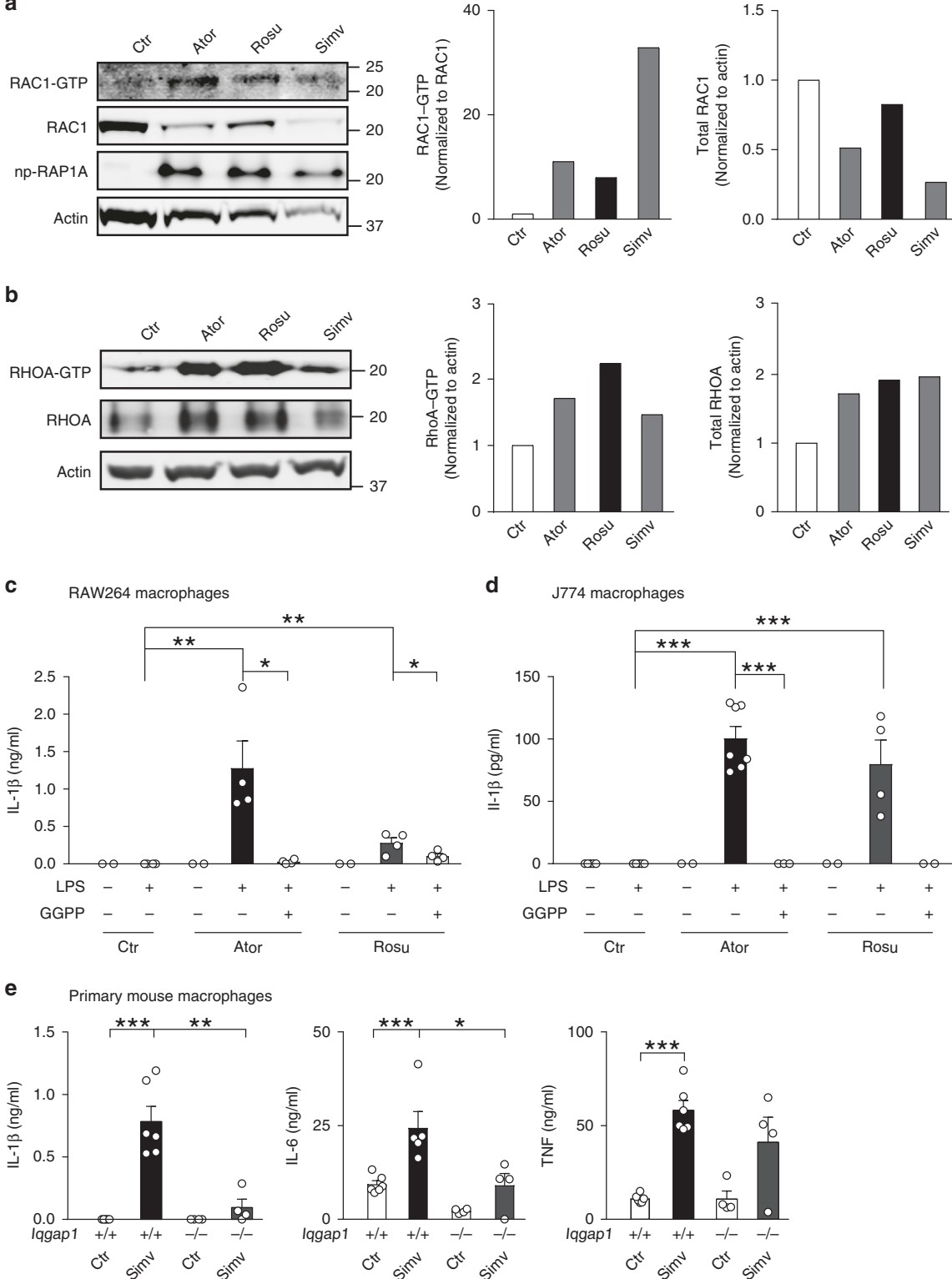

**Fig. 6** Statins increase Rac1-GTP and cytokine production in a GGPP- and Iqgap1-dependent fashion. **a** Left, western blots showing levels of GTP-bound and total Rac1 in lysates of RAW 264.7 macrophages incubated for 3 weeks with Atorvastatin (5 μM), Rosuvastatin (2.5 μM), and Simvastatin (1 μM). Np-Rap1A was used as a marker of GGTase-I-deficient cells and Actin as a loading control. Middle and right, amounts of GTP-bound and total Rac1 determined by densitometry. **b** Left, western blots showing levels of GTP-bound and total RhoA in the same cells as in (**a**). Middle and right, amounts of GTP-bound and total RhoA determined by densitometry. **c** Il-1β concentration, before and 8 h after LPS stimulation, in medium of RAW 264.7 macrophages incubated with Atorvastatin (5 μM) and Rosuvastatin (2.5 μM) for 21 days. GGPP (10 μM) was added to the cells 3 days before LPS stimulation. **d** Similar experiment as in **c** performed with J774 macrophages. **e** Cytokine concentration in medium of LPS-stimulated Iqgap1[+/+] and Iqgap1[−/−] macrophages incubated with Simvastatin (5 μM) for 60 h. Error bars represent s.e.m. Significance between groups were calculated with two tailed Student's t test. *P < 0.05, **P < 0.01, ***P < 0.001

the cell such as during proliferation or certain types of cell movement. These issues will be possible to address in the future.

But based on the current study, we conclude that a major role of GGTase-I-mediated prenylation in macrophages is to limit Rac1 activation and pro-inflammatory signaling, prevent Rac1-dependent Il-1β maturation, and thereby restrain innate immune responses.

## Methods

**Mouse breeding and genotyping.** Mice harboring conditional knockout alleles of the beta subunit of GGTase-I (*Pggt1b*fl/fl) were bred with lysozyme M-Cre (LC) knock-in mice to produce offspring lacking GGTase-I in macrophages as described[6]; these mice were designated *Pggt1b*Δ/Δ (Δ = delta, deleted allele). *Pggt1b*Δ/Δ mice were bred with mice harboring conditional knockout alleles of *Rac1*, *Rhoa*, and *Cdc42*, and conventional knockout alleles for *Iqgap1*[33,53–55]. Genotyping was performed by polymerase chain reaction (PCR) on genomic DNA from ear or tail biopsies[6]. Mice were housed in a specific pathogen-free facility monitored by routine testing of sentinel mice, and were given free access to water and chow. Mice were on a mixed genetic background (129Ola/Hsd-C57BL/6) and control mice in all experiments were gender-matched littermates. Animal experiments were approved by the research animal ethics committee in Gothenburg, Sweden.

**Histology and quantification of arthritis in joints.** Joints from 12-week-old mice were fixed in 4% paraformaldehyde and stored in 70% ethanol. The joints were decalcified, embedded in paraffin, sectioned, and stained with hematoxylin and eosin. The slides were analyzed in a Zeiss Axioplan 2 microscope (Carl Zeiss AG). Synovitis and erosion scores were evaluated in sections of knee, ankle, metatarsal, elbow, wrist, and metacarpal joints by an observer blinded to genotype. An arbitrary scale from 0 to 3 was used: 0—is a healthy joint, 1—hypertrophy/mild proliferation of synovia constituting more than two intimal lining cell layers, and mononuclear cell infiltration is visible; 2—hypercellularity, multiple inflammatory foci in the sub-lining layer, and proliferation of synovia; and 3—massive influx of inflammatory cells scattered throughout the synovial tissue, and growth of granulation tissue (pannus). Cartilage and bone erosion were evaluated with a separate 0–3 scale: 0 is a healthy joint; 1—reduction or uneven cartilage thickness; 2—compromised cartilage integrity and formation of erosions on the cartilage surface under the growing pannus; and 3—loss of articular surface, obliteration of the joint cavity, and loss of joint or bone shape. The cumulative arthritis index for each mouse was constructed by adding the scores from the fore and hind paws[56].

**Isolating and culturing bone marrow macrophages.** Bone marrow cells were isolated from femur and tibia and cultured for 7–10 days in DMEM high glucose medium (61,965,059) supplemented with 10% fetal bovine serum (10270-106, Thermo Fisher Scientific), 1% HEPES (H0887, Sigma-Aldrich), 1% glutamine (929070, Thermo Fisher Scientific), 1% gentamycin (11482524, Fischer Scientific), 0.01% β-mercaptoethanol, and 10% CMG14-12 cell supernatant as a source of macrophage colony-stimulating factor (M-CSF)[57].

**Protein analyses.** Rac1-GTP levels were assessed with the Active Rac1 Pulldown and Detection kit (16118, Thermo Fisher Scientific); Cdc42-GTP with the Active Cdc42 Pulldown and Detection kit (16119, Thermo Fisher Scientific); and RhoA-GTP with the RhoA Activation Assay Biochem Kit (BK036, Cytoskeleton). For IP, cells were lysed in a buffer containing 25 mM Tris-HCl pH 7.4, 150 mM NaCl, 1 mM EDTA, 1% NP-40, and 5% glycerol; and IP was performed with antibodies recognizing Rac1 (05-389, Millipore); Cdc42 (sc-87), Iqgap1 (sc-10792), and Tiam1 (sc-376021, Santa Cruz Biotechnology); and RhoA (ARH03, Cytoskeleton), using the Dynabeads Protein G Immunoprecipitation Kit (10007D, Thermo Fisher Scientific). Ubiquitinated Rac1 was isolated with the Pierce Ubiquitin Enrichment Kit (89899, Thermo Fisher Scientific). Subcellular fractions were isolated with the Qproteome Cell Compartment Kit (37502, Qiagen). Detergent and aqueous fractions were isolated with TX-114 assays as described[58]. To quantify Rac1 turnover rates, cell lysates were isolated from macrophages after incubation with cycloheximide (15 μg/ml) to stop protein synthesis. Western blots were performed by loading equal amounts of total proteins from whole-cell lysates or cellular fractions on Bolt 4–12% Bis–Tris gels (Thermo Fisher Scientific) and 18% sodium dodecyl sulphate polyacrylamide gel electrophoresis. Proteins were transferred to nitrocellulose membranes which were incubated with antibodies to Rac1 (05-389, Millipore); ACTIN (A1978, Sigma-Aldrich); Mmp13 (sc-30073), nonprenylated RAP1A (sc-1482), RacGAP1 (sc-98617, Santa Cruz Biotechnology); Iqgap1 (SC-376021, Santa Cruz Biotechnology), Tiam1 (A300-099A, Bethyl Laboratories); Histone-H3 (ab18521, Abcam); RhoA (2117S), Cdc42 (2462S), phospho-p38 (9211S), p38 (9212S), phospho-Stat3Y705 (9145S), Stat3 (9132S), phospho-SrcY416 (2101S), Src (2109S), phospho-Ikkα/βS176/180 (2697S), Iqgap1 (2293S), VAV1 (2502S), VAV2 (2848S), DOCK180 (4846S), Integrin-α5 (4705S), and β-PIX (4515S, Cell Signaling Technology); Il-1β (AF-401-NA, R&D System); and anti-caspase-1 (p20) (AG-20B-0042-C100, Adipogen). Protein bands were visualized with infrared dye-conjugated secondary anti-mouse (926–32212), anti-rabbit

(926–32,211), and anti-goat (926–32,214, LI-COR) antibodies and analyzed in a LI-COR Odyssey Imager. Band densities were analyzed with Image J. Mmp13 in supernatants of LPS-stimulated macrophages was analyzed by western blots using Mmp13 antibodies (sc-30073, Santa Cruz Biotechnology) and horseradish-conjugated anti-rabbit (NA934, GE Healthcare Lifesciences) antibodies and the ECL western blotting system (RPN2232, GE Healthcare Lifesciences), as described[59]. Primary antibodies were used at 1:500 dilution, except anti-Mmp13 and -TIAM1 which were used at 1:250 dilution.

**In vitro prenylation assay.** Rac1 prenylation was detected with a click chemistry approach[60]. Macrophages were incubated for 48 h with 30 μM Click-IT Geranylgeranyl Alcohol, Azide, mixed isomers (C10249, Thermo Fisher Scientific). Cells were then lysed in a buffer containing 25 mM Tris-HCl, 150 mM NaCl, 1 mM EDTA, 1% NP-40, and 5% glycerol, supplemented with protease and phosphatase inhibitors. IP was performed with Rac1 antibodies (05-389, Millipore) using the Dynabeads Protein G Immunoprecipitation Kit (10007D, Thermo Fisher Scientific). The click chemistry reaction was performed on the immunoprecipitate with a buffer containing 10 μM Alexa Fluor 488-alkyne (A10267, Thermo Fisher Scientific), 1 mM tris(2-carboxyethyl)phosphine (TCEP), 100 μM tris[(1-benzyl-1H-1,2,3-triazol-4-yl)methyl]amine (TBTA), and 1 mM CuSO4 in PBS for 1 h. The immunoprecipitates were washed three times with PBS containing 1% NP-40; eluted in LDS sample buffer; and then the proteins were resolved on 4–12% gels. Fluorescent (i.e., prenylated) Rac1 in gels was detected by Gel Doc XR+ molecular imager (BioRad).

**Rac1 ubiquitination.** Ubiquitinated Rac1 was detected as described[61]. HEK293 cells were lysed in a buffer containing 2% sodium deoxycholate, 150 mM NaCl, 10 mM Tris-HCl, and supplemented with 10 μM MG132, 10 μM PR619, protease and phosphatase inhibitors. Lysates were boiled for 10 min; sonicated; diluted ten times in a buffer containing 10 mM Tris-HCl, 150 mM NaCl, 2 mM EDTA, and 1% Triton; incubated at +4 °C for 1 h with continuous rotation; and then clarified by centrifugation at 20,000×*g* for 30 min. IP was performed with Rac1 antibodies (05-389, Millipore) using the Dynabeads Protein G Immunoprecipitation Kit (10007D, Thermo Fisher Scientific). Western blots were performed on Bolt 4–12% Bis–Tris gels (Thermo Fisher Scientific). Proteins were transferred to nitrocellulose for western blots with Ubiquitin antibodies (Thermo Fischer Scientific).

**Gene-expression analyses.** RNA was isolated from macrophages with the RNeasy Mini kit (74104, Qiagen) 8 h after LPS stimulation (tlrl-eblps, Invivogen). Complementary (c) DNA was synthesized from RNA with the iScript cDNA synthesis kit (1708890, Biorad). Quantitative real-time PCR was performed with Taqman assays for mouse *Rac1* (Mm01201657_g1), *Rhoa* (Mm00834507_g1), *Cdc42* (Mm01194005_g1), *Tiam1* (Mm00437079_m1), and *Actb* (4352933E, Thermo Fisher Scientific); and with SYBR-green using primers listed in Supplementary Table 1; in a 7900HT-fast machine (Applied Biosystems).

**Cytokine analyses and inhibitors.** Macrophages were cultured overnight in medium without M-CSF and stimulated with LPS (10 ng/ml) in fresh medium. In some experiments the macrophages were incubated for 30 min with a p38 inhibitor (SB203580, Invivogen) and a ROS inhibitor (D2926, Sigma-Aldrich) before LPS stimulation; in other experiments the macrophages were incubated for 24–48 h with small-interfering (si) RNAs targeting *Arhgdia* (AM16706, Thermo Fischer Scietific), *Tiam1* (D-047808-03-0050), *Iqgap1* (D-040589-01-0050), or containing a scrambled sequence (D-001206-14-50, Dharmacon) before LPS stimulation. For statin experiments, macrophages were treated for 60 h or 18 days with Atorvastatin, (PZ0001, Sigma Aldrich), Rosuvastatin (SML1264, Sigma Aldrich) and Simvastatin (S6196, Sigma Aldrich). GGPP ammonium salt (G6025, Sigma Aldrich) was added to the cells 3 days before LPS stimulation. Supernatants were collected before and 8 h after LPS stimulation and levels of Tnf, Il-6, and Il-1β were determined by ELISA (88-7324-76, 88-7064-76, and 88-7013-76, respectively, eBioscience).

**Mass spectrometry analysis of Rac1-interacting proteins.** Macrophages were lysed in buffer containing 25 mM Tris-HCl, 150 mM NaCl, 1 mM EDTA, 5% glycerin, and 1% CHAPS. Rac1 was immunoprecipitated with the 05-389 antibody (Millipore) using the Dynabeads Protein G Immunoprecipitation Kit (10007D, Thermo Fisher Scientific). The protein complex was eluted with 50 mM triethyl-lammonium bicarbonate and 4% SDS, and digested with trypsin using the filter-aided sample preparation method, as described[30]. The digested peptides were labeled with 10-plex isobaric tandem mass tag (TMT) reagents (Thermo Scientific). The labeled samples were combined into one TMT-set; purified with trifluoroacetic acid precipitation and HiPPR Detergent Removal Resin (Thermo Scientific). The purified sample was fractionated into twelve fractions using the Pierce High pH Reversed-Phase Peptide Fractionation Kit (Thermo Scientific), and the fractions were dried in a vacuum centrifuge and reconstituted in 20 μl of 3% acetonitrile, 0.1% formic acid for analysis. Peptides were then injected onto an Acclaim Pepmap 100 C18 trap column (2 cm × 100 μm, particle size 5 μm, Thermo Fischer Scientific) using an Easy-nano-LC 1000 liquid chromatography system and separated with an acetonitrile gradient on a 75-μm Reprosil-Pur C18-AQ column with a 300 nl/min

flow rate. Fractions were analyzed on Orbitrap Fusion Tribrid mass spectrometer (Thermo Fisher Scientific). Precursor ion mass spectra were acquired at 120.000 resolution and mass spectrometry (MS)/MS analysis was performed in a data-dependent multinotch mode where collision-induced dissociation spectra of the most intense precursor ions were recorded in ion trap at collision energy setting of 30 for 3 s. Charge states 2–7 were selected for fragmentation, dynamic exclusion was set to 30 s. MS[3] spectra for reporter ion quantitation were recorded at 60,000 resolution with higher energy collisional dissociation fragmentation at collision energy of 55 using synchronous precursor selection. In a second nano-LCMS analysis a list containing theoretical peptides from Iqgap1 and Tiam1 were included in the analysis.

The data files for the set were merged for identification and relative quantification using Proteome Discoverer version 1.4 (Thermo Fisher Scientific). The search was against the *Mus musculus* Swissprot Database version November 2017 (Swiss Institute of Bioinformatics, Switzerland) using Mascot 2.5 (Matrix Science) as a search engine with precursor mass tolerance of 5 ppm and fragment mass tolerance of 0.5 Da. Tryptic peptides were accepted with zero missed cleavage and variable modifications of methionine oxidation, fixed cysteine alkylation, and TMT-label modifications of N-termini and lysines were selected. The sum of the control samples was used as denominator and for calculating ratios. The detected peptide threshold in the software was set to a minimum quantification threshold value of 1000 and a 1% false discovery rate by searching against a reversed database and grouping identified proteins by shared sequences to minimize redundancy. Only peptides unique for a given protein were considered for identification; peptides common to other isoforms or proteins of the same family were excluded.

**F-actin staining**. Macrophages were fixed with ice-cold methanol in chamber slides and stained with Alexa Fluor 488 Phalloidin (A12379, Thermo Fischer Scientific) for 30 min. The slides were mounted with Prolong Gold Antifade Mounting reagent with DAPI (P36935, Thermo Fischer Scientific) and analyzed with confocal microscopy (LSM700, Zeiss).

**Generating RhoGDI1-knockout macrophages**. Recombinant CAS9 was purified from BL21 *Escherichia coli* (C600003, ThermoFisher Scientific) transduced with a plasmid encoding *Streptococcus pyogenes* CAS9 (gift from Niels Geijsen, Addgene plasmid #62731). Synthesized crRNA:tracrRNA (Integrated DNA Technologies) targeting mouse *Arghdia* exon 2 (sg1: CAGAUAGCUGCAGAGAAUG, sg2: CUGCGCAAGCUGCUCAGCAG; retrieved from benchling.com) were pre-incubated with CAS9 in PBS for 10 min to create readily transfectable ribonucleic proteins (RNP). Immortalized macrophages[45] were transfected with the RNPs using Lipofectamine RNAiMAX (13778150, ThermoFisher Scientific) during 48 h, and pools of cells were used for analyses.

**Generating Rac1-*CAAX* mutants with CRISPR/CAS9 editing**. We used a single plasmid approach to edit the *CAAX* sequence of Rac1. A CRISPR (cr) RNA template oligonucleotide 5′-GCTGAGACATTTACAACAGC-3′ and its complementary fragment targeting exon 6 of *Rac1* were annealed and cloned into the GeneArt-CRISPR Nuclease vector (Life Technology) downstream of a human U6 promoter. A ~1-kb DNA fragment for homologous recombination–based editing was synthesized and cloned into unique *MfeI* and *SpeI* restriction sites of the vector; the fragment was composed of a 15–18-bp sequence encoding the mutant *CAAX*-motif (i.e., 5′-CGAAAGAGAAAATCTTTA-3′ for *C-L-L-L* to *S-L-L-L* editing, and 5′-CGAAAGAGAAAATGC-3′ for *C-L-L-L* to *C-L-STOP* editing) and 0.5 kb flanking genomic sequences. In the resulting all-in-one gRNA-CAS9-editing plasmid the U6 promoter drives gRNA transcription; a cytomegalovirus promoter drives bicistronic expression of CAS9 and orange fluorescent protein linked by the 2A self-cleaving peptide; and the donor DNA sequence can replace the endogenous sequence by homologous recombination after CAS9 cleavage.

The vector was transiently transfected with jetPRIME (Polyplus) into HEK cells (HEK293, ATCC CRL-1573), and transfected single cells were sorted by fluorescence-activated cell sorting and plated in individual wells of 96-well plates. After 2–4 weeks, clones were expanded for cryopreservation and genomic DNA extraction (DNeasy blood & tissue kit, Qiagen). Knock-in events were first detected by mutation-specific PCR with forward primer 5′-CTGTCCCAACACTCCCATCAT-3′ (binding genomic DNA upstream of the 5′-homology arm) and reverse primers 5′-AACAGTAAAGATTTTCTCTTTCA-3′ (to detect the -*SLLL* editing) or 5′-TTACAAGCATTTTCTCTTTCG-3′ (to detect -*CL* editing). Second, another PCR product from potential targets—amplified with forward oligo 5′-GTGGTCGTGTTTCCTGTAGGT-3′ and reverse oligo 5′-AGTTCAGTGCTCGGTGTTCTC-3′—was TA-cloned and the plasmid of 10–12 transformed bacterial colonies for each potential target cell line was sequenced. And third, total RNA was isolated from the cells and a cDNA fragment was amplified with forward oligo 5′-CAAGTGTGTGGTGGTGGGAGA-3′ and reverse oligo 5′-AACGAGGGGCTGAGACATTTA-3′, and then sequenced using the forward oligo. Subsequently, three *CAAX* mutant (CM) cell lines were selected for experiments: *Rac1CM1*, *-2*, and *-3*. Based on the DNA and cDNA sequencing, *Rac1CM1* was predicted to have a *Rac1*$^{SLLL/CLL}$ genotype; CM2, *Rac1*$^{CL/CL}$; and CM3, *Rac1*$^{CL/−}$. A *Rac1Ctr* clone, was used as a manipulated *Rac1*$^{+/+}$ control.

**Statistics**. Values are mean and SEM unless stated otherwise. Differences between groups were determined with Student's *t* test or one-way ANOVA with Tukey's post hoc test, and were considered significant when $P < 0.05$.

**Reporting summary**. Further information on research design is available in the Nature Research Reporting Summary linked to this article.

## Data availability
The authors declare that the data supporting the findings of this study are available within the Article, supplementary information files and Source data, or are available upon reasonable requests to the authors.

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

## Acknowledgements

This study was supported by grants from the Knut and Alice Wallenberg Foundation, Strategic Research Program in Cancer at Karolinska Institutet, Center for Innovative Medicine (CIMED), Swedish Cancer Society, and Swedish Heart & Lung Foundation (to M.O.B.); the Swedish Research Council (to M.O.B, V.I.S, and M.B.); the Swedish Society for Medical Research, Knut and Alice Wallenberg Foundation, and Wallenberg Centre for Molecular and Translational Medicine (to V.I.S) and the Swedish Rheumatism Association (to M.B.). The Proteomics Core Facility at the Sahlgrenska Academy, Gothenburg University, performed the mass spectrometry analyses. Open access funding provided by University of Gothenburg.

## Author contributions

M.K.A. designed the study, performed experiments, interpreted data, made figures and statistics, and wrote the first draft of the manuscript; M.X.I. designed and performed experiments, and prepared figures; E.G.I. designed and performed the experiments, and prepared the figures; O.M.K. designed and performed the experiments; I.T.K. performed the experiments; M.E. performed the histology; C.K. performed the histology and provided technical assistance; X.X. performed the CRISPR/CAS9 gene editing; M.Br. designed the experiments; C.B. provided the mouse models; M.Bo. designed the experiments and quantified arthritis; D.W. designed the experiments and provided the reagents; V.I.S. interpreted data and provided funding; and M.O.B. designed and supervised the study, provided funding, and wrote the paper.

## Additional information

**Competing interests:** The authors declare no competing interests.

