## [Peer Review File · Nature Communications]

Reviewers' comments:

Reviewer #1 (Lipidation, G-protein signalling)(Remarks to the Author):

This study defines unique functions of non-prenylated Rac1 in macrophages that promote inflammatory responses. The study also defines similar molecular interactions of non-prenylated Rac1 in response to treatment with statins. Perhaps most importantly, the findings provide a paradigm shift by demonstrating that non-prenylated Rac1 has a biological role in inflammation. The general dogma in the field is that prenylated GTPases are the only biologically relevant form, and non-prenylated GTPases have negligible roles in biological processes. The findings reported in this manuscript dispel this dogma by defining a molecular pathway involving non-prenylated Rac1 and IQGAP1 in inflammation. The conclusions are based on several lines of experimental investigation involving the use of *Pggt1b*^{-/-} mice expressing different levels of Rac1, RhoA, or Cdc42, as well as cell lines edited by CRISPR/Cas9 to generate expression of Rac1 with an altered CAAX sequence to prevent prenylation. The results support an unexpected role for non-prenylated GTPases in inflammation and perhaps in cellular responses to statins. In general, the study is well-designed and provides novel and comprehensive results. The following issues should be addressed:

1) The authors do not provide formal evidence that Rac1, RhoA, and Cdc42 are not prenylated in the *Pggt1b*^{-/-} mice. Instead, they demonstrate that another GTPase (Rap1) is not prenylated, supporting the expected loss of geranylgeranyl transferase in the *Pggt1b*^{-/-} mice. There is the remote possibility that Rac1, RhoA, and/or Cdc42 become abnormally farnesylated in the *Pggt1b*^{-/-} mice, based on predictions supplied by the prenylation prediction suite (PrePS). This site indicates a very low probability of farnesylation for Rac1 and RhoA, but a higher probability for Cdc42 farnesylation. Although farnesylation could be viewed as a potentially negligible possibility, more formal tests of prenylation should be included. For example, a demonstration that Rac1, RhoA, and Cdc42 partition into the aqueous phase after TX-114 fractionation of cells from *Pggt1b*^{-/-} mice, or the use of click chemistry to detect reduced prenylation of the GTPases with GGPP or FPP analogs, would provide formal proof (of the likely probability) that these GTPases are not prenylated in the *Pggt1b*^{-/-} cells.

2) The diagram of the proposed model (Supplemental Figure 8) suggests that ROS generation is induced by non-prenylated Rac1 associating with IQGAP1 and TIAM, and that this ROS generation is responsible for downstream activation of STAT, p38, and Src. The prominent placement of ROS in this diagram seems premature, since the only results supporting the involvement of ROS are the data demonstrating that DPI diminishes IL1-beta production (Fig. 1G). The authors did not test whether DPI diminishes the ability of non-prenylated Rac1 to associate with IQGAP1 and TIAM (which would place ROS upstream of the complex), or whether DPI diminishes phosphorylation of STAT, p38 and Src. Without testing the effects of DPI on these events (or testing ROS generation), the diagram of the model should be modified to replace ROS with the term "Signal 1", and perhaps including the term "ROS?" in association with "Signal 1".

3) The authors should provide a brief discussion of how the dose of statins used in the cell cultures (Fig. 6) compares to physiological concentrations of statins in patients treated with these drugs. There is an ongoing controversy about whether or not the doses of statins prescribed to patients can reach high enough levels to suppress prenylation, aside from the cholesterol-lowering effects of the drugs. This discussion is particularly important since the effects of statins in patients are discussed in the fifth paragraph of the Discussion.

4) It is interesting that knockdown of Cdc42 in the *Pggt1b*^{-/-} mice increases the scores for synovitis and erosion (Supplemental Fig. 1), and Cdc42 forms more stable complexes with IQGAP1 than does Rac1 (Supplemental Fig. 5B versus Fig. 3A). It is possible that non-prenylated Cdc42 and non-

prenylated Rac1 compete for the GRD domain of IQGAP1. If this is the case, knockdown of non-prenylated Cdc42 might allow more binding of non-prenylated Rac1 to IQGAP1, which could increase Rac1/IQGAP1-dependent synovitis and erosion. Although it is not required for the proposed model, it might be interesting to determine whether the co-precipitation of Rac1 with IQGAP1 is greater in cells from Pgg1b^{-/-},Cdc42^{+/-} mice (which have diminished Cdc42 expression) compared to cells from Pgg1b^{-/-} mice. Such a finding might provide an intriguing possible explanation for the increase in synovitis and erosion when Cdc42 expression is diminished in the Pgg1b^{-/-} mice.

5) The backgrounds of some of the immunoblots are so dark that it is difficult to see some of the immunoreactive proteins in the blots (e.g., Src in Fig. 3E and RhoGDI in Fig. S4B). Lighter exposures should be presented to allow better detection of the immunoreactive proteins.

6) A more detailed description of the statistical analysis is needed. For example, it is not clear how error bars were generated in the graph shown in Figure 1A, since the figure legend states that each value was generated from n = 2. Error bars indicating SEM or SD values should be generated from n = 3 or higher.

7) Some of the opening statements in the Discussion should be modified to avoid overstating the conclusions. For example, it is stated that among the 60 proteins that are prenylated by GGTase-1, Rac1 is the only one that mediates the robust immune responses in GGTase-1-deficient mice. This is an overstatement, since all GGTase-1 substrates were not examined.

Reviewer #2 (Inflammation, PTM, immune signalling)(Remarks to the Author):

The manuscript by Akula et al addresses the mechanism by which GGTase-I-mediated prenylation regulates proinflammatory signaling in macrophages. The authors have previously shown that GGTase-I knockout increases GTP-loading of RHO family proteins, causing activation of p38 and NF-κB signaling and aberrant production of proinflammatory cytokines. Furthermore, mice lacking GGTase-I in macrophages develop severe joint inflammation resembling erosive rheumatoid arthritis. In the present study, they found that deletion of one allele of Rac1, but not Rhoa or Cdc42, prevents proinflammatory signaling in GGTase-I deficient macrophages. Depletion of GGTase-I in macrophages increases non-prenylated RAC1, which is hyperactivated through enhanced interaction with IQGAP1 and TIAM1. The hyperactivated RAC1 is responsible for the excessive production of inflammatory cytokines via activation of inflammasome and ROS-p38-NF-κB signaling pathways. These findings are interesting and provide novel insight into the function of GGTase-I in regulating proinflammatory signaling of macrophages. However, a major weakness of the manuscript is the lack of in depth investigation of the mechanism underlying the described phenotypes of the mutant mice.

Major Points:

1. In Fig. 1, how does GGTase-I and RAC1 regulate p38 and IKK activation? What are the upstream signaling factors connecting RAC1 to p38 and IKK? How does RAC1 regulate ROS production?
2. It is interesting that GGTase-I deficiency causes basal phosphorylation of STAT3 Y705, which is blocked by Rac1 deletion (Fig. 1F). The authors should discuss the potential mechanism and functional significance in proinflammatory cytokine regulation.
3. Fig. 2: How does GGTase-I regulates RAC1 ubiquitination? Does mutation of CAAX sequence of RAC1 also promotes its ubiquitination and proteolysis?
4. Fig. 5: the finding that GGTase-I regulates RAC1-IQGAP1 binding is interesting; however, how GGTase regulates RAC1-IQGAP1 interaction was not investigated. Does CAAX mutation also promote RAC1 binding to IQGAP1? The functional significance of this molecular interaction was also not studied

(e.g. via generating an interaction-defective RAC1 mutant).

5. The authors stated that non-prenylated RAC1 exhibited a reduced electrophoretic mobility compared with prenylated one (as seen in Fig. 5B). However, the more slowly migrating RAC1 band was not detected in GGTase-I deficient macrophages (Fig. 1A and 2A). Is it possible to distinguish the prenylated and non-prenylated RAC1 in immunoblot assay?

6. In Fig.2A and 2C, the authors demonstrated that RAC1-GTP levels were increased, and the total RAC1 was reduced due to ubiquitin mediated proteasomal degradation. The author did not exclude the possibility that the reduced level of total RAC1 was due to the transformation from RAC1 into GTP bound RAC1. The authors should also check whether MG132 alters GTP-bound RAC1 levels.

7. The quality of RAC1 ubiquitination assay (Fig. 2E) is low. It is odd that the ubiquitinated RAC1 is uniformly conjugated with 3 ubiquitin molecules. To confirm that this is indeed the case, the authors should repeat the experiment using a different approach. They could IP RAC1 and detect ubiquitinated RAC1 with anti-ubiquitin immunoblot.

8. IQGAP1 deletion has no effect on cytokine induction, although the loss of IQGAP1 reduced cytokine production in GGTase-I-deficient cells. This suggests that IQGAP1 is not important for RAC1 regulation under normal conditions. The authors need to provide more discussions about these findings. Authors should also examine the levels of RAC1-GTP and total RAC1 in IQGAP1 knockout macrophages.

9. In Fig.4, the authors show that IQGAP1 and TIAM1 regulate RAC1 levels in GGTase-I deficient cells. Do they regulate RAC1 ubiquitination?

Minor points:

1. In Fig2.F, the summarized changes of total RAC1 levels lack statistical analysis.

2. Figure legends should be more informative. For example, what is the difference between lanes 1,2 and 3,4 in Fig. E (left panel)?

Point-by-point response to Reviewers' comments

Reviewer #1 (Lipidation, G-protein signalling) (Remarks to the Author):

The Reviewer commented that “This study defines unique functions of non-prenylated Rac1 in macrophages that promote inflammatory responses.” and that “Perhaps most importantly, the findings provide a paradigm shift by demonstrating that non-prenylated Rac1 has a biological role in inflammation. The results support an unexpected role for non-prenylated GTPases in inflammation and perhaps in cellular responses to statins. In general, the study is well-designed and provides novel and comprehensive results.”

Thank you for the encouraging comments on our findings, for carefully reading our manuscript, and for suggesting multiple solid strategies to improve it. We have addressed your comments and are delighted at the outcome; and we hope you will share our enthusiasm.

The following issues should be addressed:

1) The authors do not provide formal evidence that Rac1, RhoA, and Cdc42 are not prenylated in the *Pggt1b*^{-/-} mice. Instead, they demonstrate that another GTPase (Rap1) is not prenylated, supporting the expected loss of geranylgeranyl transferase in the *Pggt1b*^{-/-} mice. There is the remote possibility that Rac1, RhoA, and/or Cdc42 become abnormally farnesylated in the *Pggt1b*^{-/-} mice, based on predictions supplied by the prenylation prediction suite (PrePS). This site indicates a very low probability of farnesylation for Rac1 and RhoA, but a higher probability for Cdc42 farnesylation. Although farnesylation could be viewed as a potentially negligible possibility, more formal tests of prenylation should be included. For example, a demonstration that Rac1, RhoA, and Cdc42 partition into the aqueous phase after TX-114 fractionation of cells from *Pggt1b*^{-/-} mice, or the use of click chemistry to detect reduced prenylation of the GTPases with GGPP or FPP analogs, would provide formal proof (of the likely probability) that these GTPases are not prenylated in the *Pggt1b*^{-/-} cells.

This is an important question and we spent lots of time assessing whether GGTase-I substrates are unprenylated in our first publication on the macrophage-specific GGTase-I knockout mice (Khan et al, *J. Clin. Invest.*, 2011). We apologize for not making this issue clearer and drawing attention to this fact in the current manuscript.

In that first publication, we performed metabolic labeling to test whether RAC1 and RHOA are unprenylated in GGTase-I-knockout cells. We harvested whole-cell extracts from wild-type and GGTase-I-knockout macrophages and incubated them with radioactively (³H) labeled GGPP and recombinant GGTase-I. We argued that if the proteins are unprenylated they should be prenylated in vitro with ³H-GGPP which could then be visualized by autoradiography. Thus, after incubating the extracts with ³H-GGPP and recombinant GGTase-I, we immunoprecipitated RAC1 and RHOA, ran gels, dried them, and exposed them to film. These experiments revealed no labeling of RAC1 and RHOA in wild-type cells—indicating that the proteins were already prenylated. However, there was robust labeling of RAC1 and RHOA in GGTase-I-knockout cells indicating that those proteins must have been unprenylated in the cells. Please see figure panel A below for this published data (Fig 3G of the earlier publication).

To further address this issue, we followed your suggestion of performing TX-114 and click chemistry. In the TX-114 assays, RAC1 appeared in both the aqueous (Aq) and detergent (Dt)

phases of wild-type cells, which is consistent with previous reports¹ ; RAC1 is likely kept in the Aq phase by RHO-GDI). However, consistent with the metabolic labeling results, RAC1 was exclusively found in the Aq phase (see Supplemental Figure 1A below and in the revised manuscript). RHOA and CDC42 were present exclusively in the Dt fraction in wild-type cells but both proteins exhibited a shift to the Aq phase in the GGTase-I-knockout cells. This shift was particularly striking for CDC42. The TX-114 experiments also confirmed the earlier finding that total RAC1 levels are reduced, and RHOA and CDC42 levels increased, in GGTase-I-knockout cells.

Importantly, whereas the vast majority of CDC42 was in the aqueous phase, a substantial proportion of RHOA remained in the membrane fraction in GGTase-I-knockout cells. However, the earlier metabolic labeling experiment revealed that RHOA is unprenylated in GGTase-I-knockout cells.

Thus, when considering both the old metabolic labeling and new TX-114 experiments, the data support the Reviewer's conclusion that "the likelihood of CDC42 farnesylation could be viewed as a potentially negligible possibility." The data also support the overall conclusion that these proteins are not prenylated in GGTase-I-deficient cells. We obviously can't completely rule out the possibility that a small proportion of the proteins undergo farnesylation; but it seems unlikely that this would contribute to the observed phenotypes.

Because we found that RAC1 is responsible for most of the inflammation observed in vivo and in vitro as a result of GGTase-I deficiency, we wanted to confirmed also with click-chemistry that RAC1 is indeed unprenylated in GGTase-I-deficient cells. Please see Supplemental Figure 1B panel below for this data.

The new data on TX-114 and click chemistry are presented in Supplemental Figure 1A and B in the revised manuscript; and new text is added to the first paragraph of the Results on page 5.

A In-vitro prenylation Assay (Published data, Khan et al 2011)

Sup. Fig. 1A

Sup. Fig. 1B

2) The diagram of the proposed model (Supplemental Figure 8) suggests that ROS generation is induced by non-prenylated Rac1 associating with IQGAP1 and TIAM, and that this ROS generation is responsible for downstream activation of STAT, p38, and Src. The prominent placement of ROS in this diagram seems premature, since the only results supporting the involvement of ROS are the data demonstrating that DPI diminishes IL1-beta production (Fig. 1G). The authors did not test whether DPI diminishes the ability of non-prenylated Rac1 to associate with IQGAP1 and TIAM (which would place ROS upstream of the complex), or whether DPI diminishes phosphorylation of STAT, p38 and Src. Without testing the effects of DPI on these events (or testing ROS generation), the diagram of the model should be modified to replace ROS with the term “Signal 1”, and perhaps including the term “ROS?” in association with “Signal 1”.

Thank you for pointing out this problem with the model. We agree that the proposed model needs to be modified, or that we need do more experiments!

We showed previously that knockout of GGTase-I increases ROS production in macrophages (see figure panel A below)². It is well known that RAC1 can serve as a subunit in the NADPH oxidase complex, which stimulates ROS production in immune cells³, and can trigger ROS-dependent activation of SRC and STAT3 (p38 is activated directly by RAC1/PAK, independently of ROS)^{4, 5, 6, 7}. The knockout of RAC1 in GGTase-I deficient macrophages in our study reduced phosphorylation of all three proteins, suggesting that RAC1 acts upstream. Furthermore, inhibiting ROS with DPI markedly reduced LPS-induced cytokine production in GGTase-I knockout macrophages. Although these results suggest that ROS mediates RAC1-

induced cytokine production, we agree with the Reviewer's assessment that these experiments do not reveal whether ROS mediates RAC1-induced SRC and STAT3 phosphorylation. We therefore decided to perform those experiments. We also analyzed p38 and IKK α/β in the same experiments.

We incubated wild-type and GGTase-I-knockout macrophages with DPI and determined the impact on basal and LPS-induced levels of phospho-STAT3, -SRC, -IKK α/β , and -p38. As expected, the DPI reduced or normalized the increased levels of phospho-STAT3, -SRC, and -IKK α/β in GGTase-I-knockout cells; but it did not reduce p38 phosphorylation. These results indicate that ROS mediates RAC1-induced phosphorylation of STAT3, SRC, and IKK α/β in GGTase-I-knockout macrophages. The results also indicate that RAC1-induced phosphorylation of p38 is independent of ROS and likely mediated by RAC1's interaction with PAK.

We repeated the experiments with N-acetylcysteine (NAC), a general antioxidant, and found similar effects. The DPI and NAC experiments are shown in Supplemental Figure 2G and H. Results text has been added in the third paragraph on page 6.

Your comment along with the new results prompted us to revise the model in Supplemental Figure 8. We hope you will agree that the data now make it appropriate to include an arrow from RAC1 to ROS (and on to STAT3/SRC/IKK) and a separate arrow from RAC1 to p38.

A Published data (Khan et al 2011)**Sup. Fig. 2G****Sup. Fig. 2H**
3) The authors should provide a brief discussion of how the dose of statins used in the cell cultures (Fig. 6) compares to physiological concentrations of statins in patients treated with these drugs. There is an ongoing controversy about whether or not the doses of statins prescribed to patients can reach high enough levels to suppress prenylation, aside from the cholesterol-lowering effects of the drugs. This discussion is particularly important since the effects of statins in patients are discussed in the fifth paragraph of the Discussion.

We agree that this should be discussed. In response to this comment we added the following text to the end of the fifth paragraph of the Discussion on page 13:

“However, these speculations should be interpreted with caution because there is little evidence that statins inhibit prenylation in vivo. Daily statin doses used by patients range from 5 to 80 mg/day, resulting in plasma concentrations of 1–15 nM⁸. The doses used in vitro in the present study (i.e., 1–5 μM) are lower than those of many other studies^{9, 10,11}, but they are likely higher than what cells in vivo are exposed to.”

4) It is interesting that knockdown of *Cdc42* in the *Pggt1b*^{-/-} mice increases the scores for synovitis and erosion (Supplemental Fig. 1), and *Cdc42* forms more stable complexes with IQGAP1 than does *Rac1* (Supplemental Fig. 5B versus Fig. 3A). It is possible that non-prenylated *Cdc42* and non-prenylated *Rac1* compete for the GRD domain of IQGAP1. If this is the case, knockdown of non-prenylated *Cdc42* might allow more binding of non-prenylated *Rac1* to IQGAP1, which could increase *Rac1*/IQGAP1-dependent synovitis and erosion. Although it is not required for the proposed model, it might be interesting to determine whether the co-precipitation of *Rac1* with IQGAP1 is greater in cells from *Pggt1b*^{-/-}, *Cdc42*^{+/-} mice (which have diminished *Cdc42* expression) compared to cells from *Pggt1b*^{-/-} mice. Such a finding might provide an intriguing possible explanation for the increase in synovitis and erosion when *Cdc42* expression is diminished in the *Pggt1b*^{-/-} mice.

Thank you for this innovative suggestion of determining whether knockdown or knockout of *Cdc42* would increase RAC1-IQGAP1 interactions. In response to this suggestion we knocked down *Cdc42* with siRNAs in GGTase-I-knockout cells, and found that this tended to increase RAC1-IQGAP1 interactions (see figure below). However, the data is not particularly convincing despite efficient *Cdc42* knockdown. We would like to repeat this experiment using *Pggt1b*^{fl/fl}LC mice with one and two *Cdc42* knockout alleles, and look forward to performing those experiments in around 6 months when we have re-derived more of those mice.

As the Reviewer noted, this line of experiments would not be required for the proposed model, and we hope you agree that it would be appropriate to address this question after publishing the current manuscript.

5) The backgrounds of some of the immunoblots are so dark that it is difficult to see some of the immunoreactive proteins in the blots (e.g., *Src* in Fig. 3E and *RhoGDI* in Fig. S4B). Lighter exposures should be presented to allow better detection of the immunoreactive proteins.

We agree that the mentioned blots Figures 3E and S4B needed lighter exposures. We have now fixed those blots and also a blot in Figure 1E.

6) A more detailed description of the statistical analysis is needed. For example, it is not clear how error bars were generated in the graph shown in Figure 1A, since the figure legend states that each value was generated from $n = 2$. Error bars indicating SEM or SD values should be generated from $n = 3$ or higher.

In the past, we have typically included SD or SEM for data points with $n \geq 2$; and performed statistical analyses when $n \geq 3$. However, in response to this comment, we removed error bars for all instances where $n < 3$ and updated the statistics section to illustrate this approach.

7) Some of the opening statements in the Discussion should be modified to avoid overstating the conclusions. For example, it is stated that among the 60 proteins that are prenylated by GGTase-1, Rac1 is the only one that mediates the robust immune responses in GGTase1-deficient mice. This is an overstatement, since all GGTase-1 substrates were not examined.

We agree and have toned down this segment of the Discussion. Specifically, we removed the word “only” which does not belong there; and added that RAC1 mediates *the majority* of the robust innate immune responses, rather than all of them.

Thank you again for your careful assessment of our manuscript and for your suggestions which we feel have improved both the quality of the data and the validity of our conclusions.

Reviewer #2 (Inflammation, PTM, immune signalling) (Remarks to the Author):

The Reviewer commented that “The manuscript by Akula et al addresses the mechanism by which GGTase-I-mediated prenylation regulates proinflammatory signaling in macrophages.” and that “These findings are interesting and provide novel insight into the function of GGTase-I in regulating proinflammatory signaling of macrophages. However, a major weakness of the manuscript is the lack of in depth investigation of the mechanism underlying the described phenotypes of the mutant mice.”

Thank you for carefully reading our manuscript and for the excellent suggestions for how to improve the manuscript. We have addressed your comments and are delighted at the outcome; and we hope you will share our enthusiasm.

Major Points:

1. In Fig. 1, how does GGTase-I and RAC1 regulate p38 and IKK activation? What are the upstream signaling factors connecting RAC1 to p38 and IKK? How does RAC1 regulate ROS production?

These are important points and we are happy to clarify and add new data, that – we hope you agree – increases the understanding of how RAC1 activates downstream signaling pathways.

We showed previously that knockout of GGTase-I increases ROS production in macrophages (see figure panel A below) ². It is well established that RAC1 can serve as a subunit in the NADPH oxidase complex, which stimulates ROS production in immune cells ³, and can trigger ROS-dependent activation of SRC and STAT3. p38 on the other hand is activated directly by RAC1/PAK ⁷. IKK phosphorylation can be triggered by both ROS-SRC and p38.

In our study, knockout of RAC1 in GGTase-I deficient macrophages reduced the phosphorylation of SRC, STAT3, IKK, and p38, suggesting that RAC1 is upstream of those proteins. Furthermore, inhibiting ROS with DPI markedly reduced LPS-induced cytokine production in GGTase-I knockout macrophages. Although these results suggest that ROS mediates RAC1-induced cytokine production, they do not reveal whether ROS mediates RAC1-induced SRC and STAT3 phosphorylation. Thus, in response to your comment, and one from Reviewer 1, we decided to address this issue. We also analyzed p38 and IKK α/β .

We incubated wild-type and GGTase-I-knockout macrophages with DPI and determined the impact on basal and LPS-induced levels of phospho-STAT3, SRC, IKK, and p38. As expected, the DPI reduced or normalized the increased levels of phospho-STAT3, -SRC, and -IKK in GGTase-I-knockout cells; but it did not reduce p38 phosphorylation. These results indicate that ROS mediates RAC1-induced phosphorylation of STAT3, SRC, and IKK in GGTase-I-knockout macrophages. The results also indicate that RAC1-induced phosphorylation of p38 is independent of ROS and likely mediated by RAC1's interaction with PAK.

We repeated the experiments with *N*-acetylcysteine (NAC), a general antioxidant, and found similar effects. The DPI and NAC experiments are shown in Supplemental Figure 2G and H (See figures below). Results text has been added in the third paragraph on page 6.

The new results also prompted us to revise the model in Supplemental Figure 8. We hope you will agree that the data now make it appropriate to place ROS downstream of RAC1 and

upstream of STAT3 and SRC. We thus include an arrow from RAC1 to ROS (and on to STAT3/SRC/IKK) and a separate arrow from RAC1 to p38.

A Published data (Khan et al 2011)

Sup. Fig. 2G

Sup. Fig. 2H

2. It is interesting that GGTase-I deficiency causes basal phosphorylation of STAT3 Y705, which is blocked by Rac1 deletion (Fig. 1F). The authors should discuss the potential mechanism and functional significance in pro-inflammatory cytokine regulation.

We agree with reviewer and included the following paragraph on page 12 of the Discussion:

“GGTase-I deficiency increased basal phosphorylation of SRC, STAT3, and IKK α/β —and to some extent p38—in a RAC1-dependent fashion; LPS-induced phosphorylation of all four proteins was also markedly increased. RAC1-induced activation of SRC, STAT3, and IKK was mediated by ROS, whereas p38 activation was likely a direct consequence of RAC1/PAK activity (Supplementary Figure 8). These findings are consistent with previous reports that RAC1 can trigger ROS production by NADPH oxidases which activates STAT3^{5, 6}; other studies, however, report a physical interaction between RAC1-GTP and STAT3¹². Interestingly, STAT3 contributes to the progression of chronic inflammation and joint destruction in mouse models of rheumatoid arthritis^{13,14}. SRC and p38 are involved in IKK-dependent activation of NF κ B which stimulates transcription of cytokines including IL-1 β ,

IL-6, and TNF α ^{15,16,17}. Importantly, knockout of one *Rac1* copy normalized RAC1-GTP levels in GGTase-I-deficient macrophages and reduced signaling of the entire pathway.”

3. Fig. 2: How does GGTase-I regulates RAC1 ubiquitination? Does mutation of CAAX sequence of RAC1 also promotes its ubiquitination and proteolysis?

In response to this comment we performed several new experiments.

In the earlier version of the manuscript, we found that non-prenylated RAC1 (np-RAC1) was degraded faster than prenylated RAC1; and the low levels of total RAC1 in GGTase-I-deficient cells was rescued by the administration of proteasome inhibitors (MG-132 and lactacystin). In new experiments, we can now report that the low levels of total RAC1 in RAC1-CAAX-mutant HEK293 cells (*CM1* and *CM2*) were increased following administration of proteasome inhibitors (see figure panels A and B below). We also observed increased levels of ubiquitinated RAC1 in the *CM1* and *CM2* cells. Taken together, our results indicate that when RAC1 is not prenylated—e.g., in GGTase-I-deficient cells or in RAC1-CAAX-mutant cells—RAC1 undergoes ubiquitin-mediated proteasomal degradation. The new data on the CAAX mutant cells indicate that GGTase-I is not involved in any other way than by prenylating RAC1. These data were added to Figure 5 (panels E–G) in the revised manuscript and the new Results text is added on page 10. This was an important result and we thank you for raising this point.

Fig. 5E

Fig. 5F

Fig. 5G

4. Fig. 5: the finding that GGTase-I regulates RAC1-IQGAP1 binding is interesting; however, how GGTase regulates RAC1-IQGAP1 interaction was not investigated. Does CAAX mutation also promote RAC1 binding to IQGAP1? The functional significance of this molecular interaction was also not studied (e.g. via generating an interaction-defective RAC1 mutant).

This comment follows nicely on the previous comment. In response, we tested if the interaction between RAC1 and IQGAP1 was increased in RAC1-CAAX-mutant cells. Indeed, this appeared to be the case (see figure below). The results strengthen the conclusion that when RAC1 is not prenylated it binds more avidly to IQGAP1. The data are in Figure 5H of the revised manuscript and the Results text on page 10.

There are at least two potential explanations for the nature of the increased interaction. First, the prenyl-group of wild-type RAC1 could interfere with IQGAP1 binding. And second, the remaining three amino acids on non-prenylated RAC1 (i.e., the -AAX, which are normally cleaved off following prenylation) could stimulate IQGAP1 binding. We have planned to address these issues in a follow-up project; and hope the Reviewer will agree with this plan.

Fig. 5H

5. The authors stated that non-prenylated RAC1 exhibited a reduced electrophoretic mobility compared with prenylated one (as seen in Fig. 5B). However, the more slowly migrating RAC1 band was not detected in GGTase-I deficient macrophages (Fig. 1A and 2A). Is it possible to distinguish the prenylated and non-prenylated RAC1 in immunoblot assay?

We have struggled with this issue for over a decade. It is indeed possible to detect a shift in electrophoretic mobility of prenylated versus non-prenylated RAC1. This requires running large amounts of protein extracts on extra-large (i.e., 30-40 cm) manually-poured gels and running the RAC1 band all the way to the bottom. It is impossible to observe the migration shift on conventional mini or midi gels regardless of buffer, gel percentage, antibody, and western blot approach (curiously, non-farnesylated HDJ2 and RAS are easily resolved on conventional gels). Please see below for a RAC1 western blot that was included in our first manuscript². Please also see Figure 5B of the manuscript for a western blot that was run to detect the migration shift of RAC1 in the *CAAX*-mutant cells.

6a. In Fig.2A and 2C, the authors demonstrated that RAC1-GTP levels were increased, and the total RAC1 was reduced due to ubiquitin mediated proteasomal degradation.

The author did not exclude the possibility that the reduced level of total RAC1 was due to the transformation from RAC1 into GTP bound RAC1.

Thank you for pointing out this issue; we are happy to clarify. To measure levels of active, GTP-bound, RAC1 we use affinity precipitation. The RAC1-binding domain of PAK1 is used to fish out GTP-bound RAC1 from a protein extract (the PAK1 domain does not bind to GDP-bound RAC1). We then run the affinity precipitate on a gel and perform a western blot with a RAC1 antibody. In parallel, we run the total protein extracts containing both GTP- and GDP-bound RAC1, and we refer to this as total RAC1. The same antibody is used to detect RAC1-GTP from the affinity-purified fraction as for detecting total RAC1 in the whole protein extract. Thus, the total RAC1 bands on the western blots is the sum of GTP- and GDP-bound RAC1 forms. It also means that transformation of GDP-bound to GTP-bound RAC1 can't be inferred from comparing the intensities of those bands. The simplest explanation is that RAC1-GTP is turned over faster than RAC1-GDP – potentially by being exposed to ubiquitin ligases through its interactions with IQGAP1 (alternatively RAC1-GDP binding to RHO-GDI might prevent interactions that trigger proteasomal degradation).

6b. The authors should also check whether MG132 alters GTP-bound RAC1 levels.

This is nevertheless a good idea. However, we would like to show experiments with lactacystin in the revised manuscript. The reason is that MG-132 administration—although it clearly inhibited proteasomal degradation and consistently increased total RAC1 levels—was associated with varying degrees of cellular stress. This stress may have influenced the impact of MG-132 on RAC1-GTP levels which were inconsistent, and we were unable to draw firm conclusions. Lactacystin on the other hand was well tolerated by macrophages and administration of this drug clearly increased RAC1-GTP levels, in addition to total RAC1 levels, in GGTase-I-knockout macrophages. The results are added in Supplemental Figure 3A and the results text is on page 7.

Sup. Fig. 3A

7. The quality of RAC1 ubiquitination assay (Fig. 2E) is low. It is odd that the ubiquitinated RAC1 is uniformly conjugated with 3 ubiquitin molecules. To confirm that this is indeed the case, the authors should repeat the experiment using a different approach. They could IP RAC1 and detect ubiquitinated RAC1 with anti-ubiquitin immunoblot.

Here we want to first mention that we agree that this is unusual but add that this result is highly reproducible. Please note that in addition to the data in Figure 2E, we show ubiquitination data in Supplemental Figures 5E. We agree with the Reviewer that it would be important to do the reverse experiment. Thus, we immunoprecipitated RAC1 from CAAX-mutant cells and performed western blots with ubiquitin antibodies. The results confirmed the previous conclusion that RAC1-Ub³ was increased in the mutant cells. The data is included in Figure 5G and the densitometry data was normalized to Actin and total RAC1.

Fig. 5G

8. IQGAP1 deletion has no effect on cytokine induction, although the loss of IQGAP1 reduced cytokine production in GGTase-I-deficient cells. This suggests that IQGAP1 is not important for RAC1 regulation under normal conditions. The authors need to provide more discussions about these findings. Authors should also examine the levels of RAC1-GTP and total RAC1 in IQGAP1 knockout macrophages.

In response to this relevant comment we first quantified RAC1-GTP and total RAC1 levels in *Iqgap1*-deficient macrophages. We found that knockout of *Iqgap1* reduced RAC1-GTP levels and also reduced the levels of total RAC1. See figure below. The data is added as Supplemental Figure 5D and the Results text are on page 8.

Sup. Fig. 5D

As suggested, we also provide a new paragraph in the Discussion (page 13) to address these results:

“The knockout of *Iqgap1* rescued most of the robust proinflammatory phenotypes of GGTase-I-deficient macrophages but it did not influence LPS-induced cytokine production of GGTase-I wild-type macrophages—despite reducing RAC1-GTP levels. The simplest explanation for these observations is that both the levels of RAC1-GTP and the affinity of RAC1 for IQGAP1 were markedly higher in GGTase-deficient than wild-type macrophages; thus IQGAP1’s role in controlling RAC1-GTP levels and its downstream signaling could simply be comparatively more important in the GGTase-I-deficient cells. Whether or not IQGAP1 influences RAC1-

GTP signaling and cytokine production in arthritis and other inflammatory diseases in the setting of wild-type GGTase-I remains to be determined.”

9. In Fig.4, the authors show that IQGAP1 and TIAM1 regulate RAC1 levels in GGTase-I deficient cells. Do they regulate RAC1 ubiquitination?

In response to this comment we performed several experiments designed to test the role of *Iqgap1* in RAC1 ubiquitination; for this we used *Iqgap1*-knockout cells and the results were reproducible and convincing in our view. However, unfortunately we ran into problems with the *Tiam1* knockdown experiments (we don't have *Tiam1* knockout mice): although these experiments have worked well in the past we did not obtain sufficient knockdown of *Tiam1* to be able to draw firm conclusions. We therefore hope the Reviewer will be satisfied with the data on IQGAP1.

The data reveal that knockout of *Iqgap1* reduced RAC1 ubiquitination in GGTase-I-deficient macrophages. This effect was associated with an expected increase in total RAC1 levels. These results indicate that IQGAP1 contributes to RAC1 ubiquitination in GGTase-I-deficient cells. The data is shown in Supplemental Figure 5E and Results text is on page 8.

Sup. Fig. 5E

Minor points:

1. In Fig2.F, the summarized changes of total RAC1 levels lack statistical analysis.

To be able to perform statistics, we performed new independent experiments and updated the figure according to your suggestion. See figure below.

Fig. 2F
2. Figure legends should be more informative. For example, what is the difference between lanes 1,2 and 3,4 in Fig. E (left panel)?

Thank you for bringing this lack of clarity to our attention. In the revised Figure 2E we have indicated in the figure that this is simply two independent experiments. Lanes 1 and 2 is Experiment 1; and Lanes 3 and 4, Experiment 2. We hope this is clear now. We have gone through the rest of the figures and legends and made a few similar adjustments.

Thank you again for your careful evaluation and excellent suggestions for improvement. We hope that you will now find our manuscript acceptable for publication.

References for the Response Letter

1. Michaelson D, *et al.* Rac1 accumulates in the nucleus during the G2 phase of the cell cycle and promotes cell division. *The Journal of cell biology* **181**, 485-496 (2008).
2. Khan OM, *et al.* Geranylgeranyltransferase type I (GGTase-I) deficiency hyperactivates macrophages and induces erosive arthritis in mice. *The Journal of clinical investigation* **121**, 628-639 (2011).
3. Abo A, Pick E, Hall A, Totty N, Teahan CG, Segal AW. Activation of the NADPH oxidase involves the small GTP-binding protein p21rac1. *Nature* **353**, 668-670 (1991).
4. Giannoni E, Buricchi F, Raugei G, Ramponi G, Chiarugi P. Intracellular reactive oxygen species activate Src tyrosine kinase during cell adhesion and anchorage-dependent cell growth. *Molecular and cellular biology* **25**, 6391-6403 (2005).
5. Liu J, *et al.* The ROS-mediated activation of IL-6/STAT3 signaling pathway is involved in the 27-hydroxycholesterol-induced cellular senescence in nerve cells. *Toxicol In Vitro* **45**, 10-18 (2017).

6. Yoon S, *et al.* STAT3 transcriptional factor activated by reactive oxygen species induces IL6 in starvation-induced autophagy of cancer cells. *Autophagy* **6**, 1125-1138 (2010).
7. Zhang S, *et al.* Rho family GTPases regulate p38 mitogen-activated protein kinase through the downstream mediator Pak1. *The Journal of biological chemistry* **270**, 23934-23936 (1995).
8. Bjorkhem-Bergman L, Lindh JD, Bergman P. What is a relevant statin concentration in cell experiments claiming pleiotropic effects? *Br J Clin Pharmacol* **72**, 164-165 (2011).
9. Youssef S, *et al.* The HMG-CoA reductase inhibitor, atorvastatin, promotes a Th2 bias and reverses paralysis in central nervous system autoimmune disease. *Nature* **420**, 78-84 (2002).
10. Chow OA, *et al.* Statins enhance formation of phagocyte extracellular traps. *Cell host & microbe* **8**, 445-454 (2010).
11. Metais C, Hughes B, Herron CE. Simvastatin increases excitability in the hippocampus via a PI3 kinase-dependent mechanism. *Neuroscience* **291**, 279-288 (2015).
12. Simon AR, Vikis HG, Stewart S, Fanburg BL, Cochran BH, Guan KL. Regulation of STAT3 by direct binding to the Rac1 GTPase. *Science* **290**, 144-147 (2000).
13. Oike T, *et al.* Stat3 as a potential therapeutic target for rheumatoid arthritis. *Scientific reports* **7**, 10965 (2017).
14. Ogura H, *et al.* Interleukin-17 promotes autoimmunity by triggering a positive-feedback loop via interleukin-6 induction. *Immunity* **29**, 628-636 (2008).
15. Yang Y, *et al.* Functional roles of p38 mitogen-activated protein kinase in macrophage-mediated inflammatory responses. *Mediators Inflamm* **2014**, 352371 (2014).
16. Irtegun S, Pektanc G, Akkurt ZM, Bozkurt M, Turkcu FM, Kalkanli-Tas S. Pharmacological Inactivation of Src Family Kinases Inhibits LPS-Induced TNF-alpha Production in PBMC of Patients with Behcet's Disease. *Mediators Inflamm* **2016**, 5414369 (2016)
17. Byeon SE, Yi YS, Oh J, Yoo BC, Hong S, Cho JY. The role of Src kinase in macrophage-mediated inflammatory responses. *Mediators Inflamm* **2012**, 512926 (2012).

REVIEWERS' COMMENTS:

Reviewer #1 (Remarks to the Author):

The authors have responded appropriately to this reviewer's comments regarding the first submission. Specifically, they have provided more formal evidence of the prenylation status of Rac1 in the *Pggt1b*^{-/-} mice, and they have included studies using DPI and NAC to clarify the roles of ROS in the proposed signaling pathway (Sup. Figs. 2E, 2F, and 8). Other concerns were also addressed, or will be appropriately investigated in future studies.

The new data continue to support the novel conclusion that non-prenylated Rac1 has an important biological function by interacting with IQGAP1 and promoting an inflammatory response in macrophages. The discovery of these unexpected interactions and functions of non-prenylated Rac1 dispels the previously held view that only prenylated GTPases are biologically active, and provides a strong rationale for future studies of the roles of non-prenylated GTPases in health and disease.

Reviewer #2 (Remarks to the Author):

The authors have addressed most of my original comments, and the revised manuscript is significantly improved. The only remaining concern I have is regarding the quality of RAC1 ubiquitination (original point 7). Authors performed the suggested experiment, but the quality of the data is not particularly strong. Since they only showed a small area (a single band) of the gel, it is hard to know whether RAC1 is also conjugated with poly ubiquitin chains. If there is technical difficulty, they should at least discuss this in the text not to completely rule out the other forms of ubiquitination.

Point-by-point response to Reviewers' comments

Reviewer 1 wrote that “The authors have responded appropriately to this reviewer's comments regarding the first submission.” And that “The new data continue to support the novel conclusion that non-prenylated Rac1 has an important biological function by interacting with IQGAP1 and promoting an inflammatory response in macrophages. The discovery of these unexpected interactions and functions of non-prenylated Rac1 dispels the previously held view that only prenylated GTPases are biologically active, and provides a strong rationale for future studies of the roles of non-prenylated GTPases in health and disease.”

Thank you for your careful review of our manuscript and for your constructive comments.

Reviewer 2 wrote that “The authors have addressed most of my original comments, and the revised manuscript is significantly improved.”

We thank you for your careful review and suggestions for experiments to improve our study.

Reviewer 2 also wrote that: “The only remaining concern I have is regarding the quality of RAC1 ubiquitination (original point 7). Authors performed the suggested experiment, but the quality of the data is not particularly strong. Since they only showed a small area (a single band) of the gel, it is hard to know whether RAC1 is also conjugated with poly ubiquitin chains. If there is technical difficulty, they should at least discuss this in the test not to completely rule out the other forms of ubiquitination.”

Thank you for this appropriate suggestion. We have now added a comment in the results that acknowledges that we can't rule out the possible existence of mono-, di-, and poly-ubiquitinated forms of RAC1. This was added to page 8 (Results) first paragraph.